# Towards Professional-Grade Financial Agents: Benchmarking, Tooling, and Structured Reasoning

Cheng Huang [* 1 2]   Jinghua Piao [* 3 2]   Ranran Wang [4 2]   Yong Li [3 2]

## Abstract

While Large Language Model (LLM) agents have shown encouraging progress in financial reasoning, their effectiveness in realistic financial workflows is severely hindered by the lack of holistic benchmarks and the fragility of unstructured reasoning. To address this gap, we introduce ProFinR, the first Professional Finance Reasoning benchmark, covering four financial domain tasks and comprising 528 expert-designed problems. To solve these complex financial reasoning questions, we construct the Financial Tool Universe, a tool library containing 53 domain-specific tools organized into 13 categories. Building on the tool library, we introduce ProFinAgent, a structured agent framework based on Directed Acyclic Graph (DAG) and Case-Based Memory (CBM). Compared with strictly sequential workflows, ProFinAgent coordinates tool execution through a DAG planner for parallel efficiency and uses CBM to retrieve prior cases for more reliable decision-making. Experimental results demonstrate that ProFinAgent achieves a 49.81% performance gain over state-of-the-art baselines with a 47.1% reduction in inference latency. Our code is available at https://github.com/tsinghua-fib-lab/ProFinAgent.

## 1. Introduction

Recent advances in large language models (LLMs) have transformed financial intelligence systems, shifting them from passive information retrieval to agentic decision support with financial reasoning capabilities. They perform demanding tasks such as investment research and market trend analysis. As a result, they have become essential tools in financial decision-making (Wu et al., 2023; Yang et al., 2023). This progress has also accelerated the development of specialized benchmarks aimed at evaluating agents' abilities, particularly in modeling market dynamics and inferring latent signals from financial data.

Despite this progress, a significant gap exists between current evaluation benchmarks and the requirements of real-world financial analysis. In practice, analysts operate over non-stationary, high-velocity data streams and execute end-to-end, multi-stage pipelines that integrate information acquisition, hypothesis formation, and iterative validation. In contrast, prevailing benchmarks such as FinQA (Chen et al., 2021) and FinEval (Guo et al., 2025) focus on static document-centric QA or single-shot numerical reasoning, offering limited coverage of workflow-level competence. Multi-modal datasets (Xie et al., 2024) expand input modalities but do not involve autonomous tool use or long-horizon planning. Moreover, even recent financial agent benchmarks (Hu et al., 2025; Guo et al., 2025; Choi et al., 2025; Bigeard et al., 2025) define agency primarily as data retrieval. Consequently, the lack of a realistic and high-fidelity financial reasoning benchmark constrains progress in agentic financial systems and exacerbates the disconnect between the benchmark and the real world.

Furthermore, deploying LLMs for high-stakes financial reasoning remains a significant risk. Native models are easily led to hallucinations, generating numerically plausible but factually incorrect analyses. While recent financial agents (Xiong et al., 2025; Yu et al., 2024; Cheng & Chin, 2024) attempt to address this issue with ReAct-based frameworks (Yao et al., 2022), they typically rely on strictly sequential execution. Such linear architectures lack the structural constraints required for professional analysis. Over long-horizon workflows, these limitations lead to error accumulation and context saturation, rendering agents unable to guarantee the

---

[*]Equal contribution [1]School of Information and Software Engineering, University of Electronic Science and Technology of China, Chengdu, China [2]Zhongguancun Academy, Beijing, China [3]Department of Electronic Engineering, Tsinghua University, Beijing, China [4]National Engineering Laboratory for Big Data Analysis and Applications, Peking University, Beijing, China. Correspondence to: Jinghua Piao <piaojh9727@gmail.com>, Ranran Wang <ranranw@pku.edu.cn>, Yong Li <liyong07@tsinghua.edu.cn>.

*Proceedings of the 43rd International Conference on Machine Learning*, Seoul, South Korea. PMLR 306, 2026. Copyright 2026 by the author(s).

*Table 1.* Comparison of Financial Benchmarks: From Static QA to End-to-End Investigation.

| Benchmark | Core Task | Data Source | Tool Use | Trajectory-Oriented Evaluation |
|---|---|---|---|---|
| FinTextQA (Chen et al., 2024) | L1 | Static Text | ✗ | ✗ |
| FinQA (Chen et al., 2021) | L1 | Static Table | ✗ | ✗ |
| FinEval (Guo et al., 2025) | L2 | Static Text | ✓ | ✗ |
| FinBen (Xie et al., 2024) | L1 | Multi-modal | ✗ | ✗ |
| FinanceBench (Islam et al., 2023) | L1 | PDF Documents | ✗ | ✗ |
| FinSearchComp (Hu et al., 2025) | L2 | Search | ✓ | ✗ |
| FinAgentBench (Choi et al., 2025) | L1 | Search | ✓ | ✗ |
| **ProFinR (Ours)** | **L3** | **Dynamic APIs + Hybrid** | **✓ (53 Domain Tools)** | **✓** |

**Note:** ✓: Fully Supported; ✗: Not Supported. **Tool Use**: Supports External Tools. **Evaluation Trajectory**: Evaluates the logical correctness of the intermediate reasoning chain. **L1**: Data Retrieval; **L2**: Data Analysis; **L3**: End-to-End Investigation.

reliability of financial analysis.

To address this limitation, we introduce ProFinR, a comprehensive benchmark for tool-based financial reasoning. ProFinR contains 528 challenging queries across four financial domains. We further construct a financial tool universe, a tool library designed to support end-to-end professional workflows. In addition, we present ProFinAgent, an integrated framework that combines a strict evaluation protocol with a structured agent design. Inspired by the workflow of professional analysts, the agent is equipped with a curated set of domain-specific tools. Since real financial analysis rarely follows a linear process, we propose a dependency-aware DAG planner that enforces topological consistency. To further emulate expert learning behavior, we develop an evolutionary reflective memory module that improves decision strategies through accumulated case experience.

Our contributions are summarized as follows: (1) We construct a Financial Tool Universe, providing a standardized and extensible foundation for tool-augmented financial analysis. (2) We establish ProFinR, the first holistic benchmark for autonomous financial analysis, which stress-tests agents across four financial domains and moves beyond binary accuracy to assess reasoning depth and workflow competence. (3) We propose ProFinAgent, a professional analysis framework that integrates domain-specific tool utilization, dependency-aware non-linear planning, and reflective memory, explicitly mitigating the efficiency bottlenecks and hallucination risks of strictly sequential baselines. (4) Experiments demonstrate that ProFinAgent substantially outperforms state-of-the-art models, delivering a 49.81% performance uplift over strong baselines while reducing inference latency by 47.1%.

**Conflict of Interest Disclosure:** The authors declare that they have no affiliated financial interests or commercial conflicts to disclose. This research was conducted purely for academic purposes, without any direct funding, employment, or influence from corporate entities.

## 2. Benchmark

In this section, we introduce ProFinR, a rigorously curated evaluation framework designed to probe the boundaries of autonomous agents within the financial domain. In contrast to prior datasets summarized in Table 1, which are primarily confined to static text comprehension, ProFinR evaluates the full spectrum of agentic cognition. ProFinR targets dynamic information acquisition, multi-step quantitative reasoning, and holistic decision-making. Moreover, ProFinR integrates three complementary dimensions from domain breadth to process completeness. Domain Breadth covers multiple asset classes and data types, including corporate fundamentals and macroeconomic indicators. Reasoning Depth organizes tasks by cognitive complexity, progressing from atomic retrieval to autonomous investigation. Process Completeness requires end-to-end workflow execution. It includes data retrieval, model optimization, and report generation.

*Table 2.* Hierarchical Examples of Financial Tasks

| Level | Task Type | Representative User Queries |
|---|---|---|
| L1 | Data Retrieval | *What was the closing price of Meta Platforms (META) on Dec. 29, 2023?* 
 *Retrieve quarterly balance sheet data for Meta Platforms (META) from Q1 2021 to Q4 2023.* |
| L2 | Data Analysis | *Review Apple's (AAPL) price movements during 2023 for volatility analysis.* 
 *Calculate the 14-day RSI for Coinbase (COIN) from 2023-01-01 to 2023-06-30.* |
| L3 | End-to-End Investigation | *Compute the average daily Rank IC of $\text{Mean}(\$close, 10)/\text{Mean}(\$close, 30) - 1$ against next-day returns for CSI 300 constituents (2021–2023).* 
 *Produce a risk assessment report for Boeing (BA) covering supply chain disruptions and safety incidents (2024–2025).* |

### 2.1. Task Classification

As shown in Table 2 and Figure 1, we organize tasks into three difficulty levels and four financial domains, covering a broad spectrum of real-world financial workflows.

We formulate a financial reasoning task as a tuple $\mathcal{X} =$

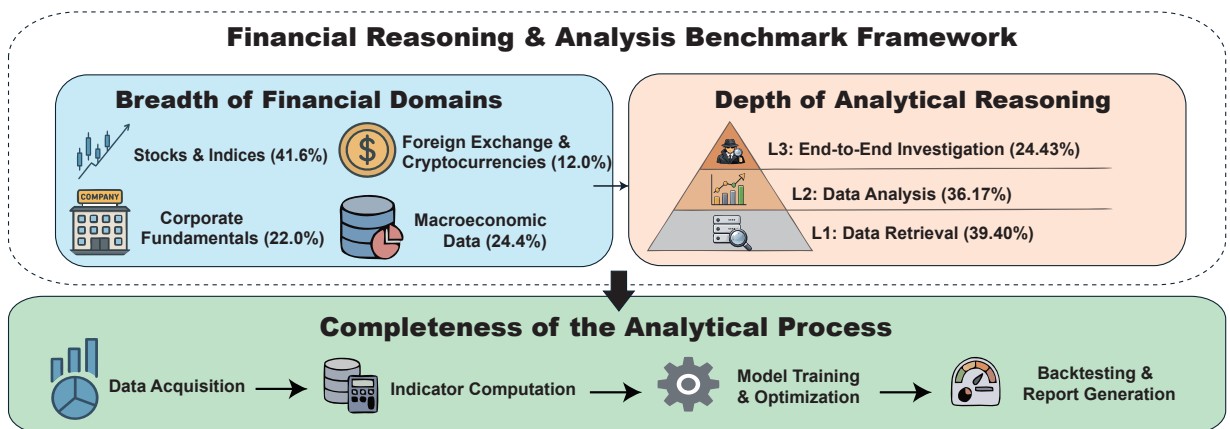

*Figure 1.* Overview of ProFinR, which evaluates agent capabilities across domain breadth, reasoning depth, and process completeness.

$(q, \mathcal{E}, y^*)$, where $q$ denotes the natural language query and $\mathcal{E}$ represents the external environment containing the accessible toolset $\mathcal{A}$ and the financial database $\mathcal{D}$. The objective is to generate a final answer $y$ that aligns with the ground truth $y^*$. We categorize tasks into three hierarchical levels based on cognitive complexity.

Level 1 (Data Retrieval) targets the precise identification of specific financial indicators conditioned on user queries, resolving semantic ambiguities in metric definitions. Advancing to Level 2 (Data Analysis), the agent must synthesize multi-source data to construct valid arithmetic computation graphs, resolving challenges in cross-document alignment. Finally, Level 3 (End-to-End Investigation) orchestrates full-cycle workflows involving strategic planning and report generation to ensure data completeness and analytical depth, strictly validating the rationality of derived insights over long horizons.

We further broaden the evaluation landscape to encompass four pivotal financial domains. These domains span diverse sectors ranging from granular technical analysis to global macroeconomic policy. This diversity evaluates agent generalization across data modalities and domain-specific terminology.

### 2.2. Multi-dimensional Evaluation Protocol

Previous exact-match metrics fail to capture the semantic depth of autonomous financial analysis. Such workflows demand rigorous validation of numerical precision and professional viability beyond simple text overlap. To bridge this gap, we establish a robust evaluation standard. We implement a two-stage verification pipeline merging automated filtering with expert auditing. An LLM-based evaluator reviews responses for format compliance. Domain experts then analyze the content to ensure factual accuracy and reduce hallucinations in high-stakes situations. Furthermore, we quantify agent performance using three orthogonal dimensions.

**Numerical Precision** ($S_{\text{val}}$) enforces a strictly binary constraint on factual accuracy. It validates whether the relative error between the final answer $y$ and ground truth $y^*$ remains within a pre-defined tolerance $\epsilon$ (1%). We define this metric as an indicator function to ensure rigorous factual groundedness:

$$S_{\text{val}} = \mathbb{I}\left(\left|\frac{y - y^*}{y^*}\right| < \epsilon\right) \tag{1}$$

**Tool Alignment** ($S_{\text{tool}}$) quantifies the procedural correctness of the execution chain based on functional categories. Let $\mathcal{T}_{\text{pred}}$ and $\mathcal{T}_{\text{gold}}$ denote the sets of tool categories derived from the agent execution and the expert trajectory via the mapping function $\phi : \mathcal{A} \to \mathcal{W}$. We compute the alignment score using the Jaccard similarity coefficient:

$$S_{\text{tool}} = \frac{|\mathcal{T}_{\text{pred}} \cap \mathcal{T}_{\text{gold}}|}{|\mathcal{T}_{\text{pred}} \cup \mathcal{T}_{\text{gold}}|} \tag{2}$$

**Soundness** ($S_{\text{sound}}$) explicitly targets the complexity of Level 3 tasks. This dimension evaluates the logical completeness of the analysis by assessing whether the generated report synthesizes retrieved evidence into a coherent investment thesis aligned with professional standards. The calculation formula can be expressed as follows:

$$S_{\text{sound}} = J_1 \times J_2 \times \frac{K_{\text{matched}}}{K_{\text{total}}} \tag{3}$$

where $J_1$ and $J_2$ denote the binary pass decisions of the two evaluating models, and the fraction represents the exact coverage of required key information.

## 3. Framework

In this section, we introduce ProFinAgent, a structure-augmented framework designed to impose deterministic

**ProFinAgent Framework**

*Figure 2.* Overview of the ProFinAgent framework, which integrates professional tools, DAG-guided planning, and evolutionary memory.

topological constraints on financial reasoning. The architecture synergizes three tightly coupled modules. The Professional Financial Tooling Layer first standardizes interactions with external environments through hierarchical retrieval. The DAG-Guided Tool Planner subsequently orchestrates execution trajectories to enforce logical dependencies and enable parallel acceleration. This workflow is reinforced by the Evolutionary Case-Based Memory Framework, which establishes a continuous feedback loop by utilizing historical experiences to refine future decision-making.

## 3.1. Architecture Overview

As illustrated in Figure 2, ProFinAgent integrates three core components to transform unstructured queries into verifiable financial outputs.

The Professional Financial Tooling Layer standardizes the interaction with external environments. It encapsulates a comprehensive universe of domain-specific instruments. To mitigate action space sparsity, the module employs hierarchical retrieval. This mechanism prunes irrelevant APIs to synthesize a compact, high-fidelity context for the planner. The DAG-Guided Tool Planner subsequently orchestrates the inference trajectory. It constructs a dependency-aware Directed Acyclic Graph to encode strict data lineage. This topological structure enables the execution engine to dispatch independent sub-tasks in parallel. The design ensures computational efficiency while maintaining rigorous logical precedence. The Evolutionary Case-Based Memory Framework closes the reasoning loop. It accumulates historical execution trajectories to form a dynamic experience repository. By retrieving validated analogues, the system explicitly guides future decision-making boundaries. This evolutionary feedback continuously refines the agent's planning policy against recurring pitfalls.

## 3.2. Professional Financial Tooling Layer

To enable precise financial reasoning, we construct a specialized action space and employ a two-stage retrieval mechanism. This coarse-to-fine strategy ensures the agent can effectively use professional financial tools while maintaining reasoning efficiency.

### 3.2.1. FINANCIAL TOOL UNIVERSE

We bridge the gap between unstructured LLM outputs and structured API execution through a unified adaptation layer. As shown in Table 7, we integrate 53 distinct financial tools across 13 functional categories. These categories cover the full spectrum of financial tasks, ranging from market data retrieval to quantitative modeling. Each tool $t \in \mathcal{A}$ is encapsulated with a rigorous JSON schema definition. This schema explicitly defines the function name, parameter types, and descriptions. Such standardization allows the agent to treat diverse external APIs as uniform executable functions. Consequently, the model interacts with high-fidelity financial data sources without hallucinating parameter structures. By enforcing schema-level validation and canonicalizing heterogeneous responses into a shared intermediate representation, the layer enables compositional multi-step toolchains with deterministic argument passing and auditable execution traces. In this way, tool usage becomes a contract-driven procedure rather than a free-form generation, substantially improving reliability and reproducibility in complex financial workflows.

### 3.2.2. HIERARCHICAL TOOL RETRIEVAL

To mitigate the combinatorial complexity of searching the full action space $\mathcal{A}$, we employ a coarse-to-fine retrieval mechanism (Algorithm 1). This process dynamically constructs a relevant toolset $\mathcal{A}^*$ through two sequential stages.

**Type Filtration.** The first stage prunes the search space via generative reasoning. Given a user query $q$, the LLM identifies the necessary functional domains $\mathcal{W}_q \subset \mathcal{W}$ from the available categories. We derive the candidate subset by aggregating all tools belonging to these selected domains:

$$\mathcal{A}_{\text{sub}} = \{t \in \mathcal{A} \mid \text{Type}(t) \in \mathcal{W}_q\}. \tag{4}$$

This step efficiently filters out categorically irrelevant tools before the system performs computationally expensive semantic analysis.

**Fine-Grained Hybrid Ranking.** The second stage synthesizes the final toolset $\mathcal{A}^*$ from $\mathcal{A}_{\text{sub}}$ using a hybrid strategy. We first apply hard constraint injection. A deterministic mapping identifies domain-critical tools based on technical keywords in $q$, forming the priority set $\mathcal{A}_{\text{rule}}$. Next, we perform dynamic semantic pruning on the remaining candidates. We compute the cosine similarity $s_i = \cos(\mathbf{e}_q, \mathbf{e}_{t_i})$ between the query and tool embeddings. Tools are retained in the semantic subset $\mathcal{A}_{\text{sem}}$ only if they satisfy two criteria: $s_i > 0.7$ and a rank within the top-$K$. The cutoff $K$ adapts dynamically to the candidate set size:

$$K = \lceil 0.5 \times |\mathcal{A}_{\text{sub}}| \rceil. \tag{5}$$

The final action space is defined as the union $\mathcal{A}^* = \mathcal{A}_{\text{rule}} \cup \mathcal{A}_{\text{sem}}$. This approach provides the planner with a concise yet high-recall set of tools robust to linguistic variability.

## 3.3. DAG-Guided Tool Planner

Following tool selection, we transition from semantic retrieval to structural planning. We transform the discrete candidate tools $\mathcal{A}^*$ into a structured execution plan, enforcing logical rigor while maximizing computational efficiency.

### 3.3.1. TRAJECTORY STRUCTURING

We leverage the LLM $\Pi_\theta$ to synthesize a directed acyclic graph $G(V, E)$ conditioned on the augmented prompt $P_{\text{ctx}}$. Vertices $V$ represent atomic tool invocations, while edges $E$ encode strict data dependencies. Unlike sequential Chain-of-Thought approaches, this graph explicitly models the topology of the financial reasoning process. To render $G$ executable, we apply a LAYEREDTOPSORT algorithm. This algorithm decomposes the graph into a sequence of discrete execution strata $\mathbb{L} = \{L_1, \ldots, L_K\}$. This stratification guarantees that for any layer $L_k$, all constituent nodes $v_i \in L_k$ exhibit mutual independence. Consequently, we identify the maximal set of sub-tasks eligible for concurrency without violating data lineage constraints.

### 3.3.2. PARALLEL ACCELERATION

The execution engine iterates through the layers from $k = 1$ to $K$, dispatching all nodes $v_i \in L_k$ to parallel threads. We enforce a dependency-locking mechanism where a downstream node remains blocked until all ancestor nodes in previous layers have terminated. During execution, input parameters are dynamically resolved from the global observation context $C$. To optimize throughput for information-heavy tasks, we incorporate the PARDISTILL operator (Algorithm 1). This mechanism activates strictly when raw tool output $o_{\text{raw}}$ violates the density constraint $\tau_{\text{len}}$. We partition the corpus into overlapping strata $\mathcal{S} = \{c_1, \ldots, c_m\}$ using a vectorized operation. We then broadcast the query embedding $\mathbf{e}_q$ across segment embeddings to impose a semantic gate. We reconstruct a high-fidelity evidence stream based on cosine similarity:

$$o_i = \|_{j=1}^{m} \{c_j \mid \cos(\mathbf{e}_q, \mathbf{e}_{c_j}) > \tau_{\text{sim}}\}. \tag{6}$$

$\|$ denotes chronological concatenation and $\tau_{\text{sim}}$ represents the signal threshold. This process filters stochastic noise, maximizing the information density of context $C$ while maintaining the speed required for real-time DAG execution.

## 3.4. Evolutionary Case-Based Memory Framework

We implement an In-Context evolutionary strategy to refine the planning policy $\Pi_\theta$ without parameter updates. This module establishes a closed-loop feedback mechanism using historical trajectories to enhance future reasoning.

### 3.4.1. EXPERIENCE ACCUMULATION AND ARCHIVING

Upon episode conclusion, the system synthesizes a structured case study utilizing the SELFREFLECT operator (Algorithm 1). The LLM generates a retrospective analysis $f_{\text{new}}$ that encapsulates the user query $q$, dependency graph $G$, execution trace $C$, and final response $y$. We archive validated plans as positive exemplars to reinforce correct tool selection and parameter formatting. Conversely, erroneous paths yielding quality $s < \tau_{\text{thres}}$ are annotated with diagnostic critiques. These serve as negative constraints to prevent the recurrence of logical fallacies. The memory $\mathcal{M}$ is continuously updated via the rule $\mathcal{M}' \leftarrow \mathcal{M} \cup \{(q, f_{\text{new}})\}$, creating a dynamic repository of domain expertise.

### 3.4.2. CASE-BASED RETRIEVAL AND REASONING

During the perception phase of a new task, the system queries the memory repository to retrieve the most pertinent analogue. We employ a strict semantic filtering protocol to ensure high-fidelity context injection:

$$m_{\text{hist}} \leftarrow \text{KNN}(q, \mathcal{M}, k = 1), \quad \cos(\mathbf{e}_q, \mathbf{e}_m) \geq 0.7. \tag{7}$$

We retrieve the single most relevant case satisfying this similarity threshold. This reference is injected into the augmented prompt $P_{\text{ctx}}$. The antecedent context directly guides the planner $\Pi_\theta$ in identifying optimal tool types, structuring the DAG topology, and grounding API parameters.

*Table 3.* Comparative Evaluation of LLMs on ProFinR. *Setting I* evaluates native zero-shot inference, while *Setting II* evaluates the ProFinAgent workflow.

| Model Setting & Name | Accuracy | | | | Total Score | Δ vs Native |
|---|---|---|---|---|---|---|
| | L1 | L2 | L3 | Avg. | | |
| *Baseline (Naive)* | | | | | | |
| Fin-R1 | 3.85% | 0.52% | 0.00% | 1.46% | 1.70% | – |
| Qwen3-4B-Instruct-2507 | 4.32% | 0.00% | 0.00% | 1.44% | 1.70% | – |
| Qwen3-32B | 5.28% | 0.00% | 0.00% | 1.76% | 2.08% | – |
| DeepSeek-V3.2 | 16.35% | 3.66% | 5.43% | 8.48% | 9.10% | – |
| DeepSeek-R1 | 17.78% | 13.08% | 29.45% | 20.10% | 18.93% | – |
| GPT-5.2 | 16.82% | 19.37% | **65.89%** | 33.96% | 29.73% | – |
| Gemini-3 Pro | 26.44% | 20.41% | 40.30% | 29.05% | 27.65% | – |
| Claude-Sonnet 4.6 | **38.94%** | **32.98%** | 58.14% | **43.35%** | **41.48%** | – |
| *Ours (ProFinAgent)* | | | | | | |
| Qwen3-4B-Instruct-2507 | 40.30% | 28.79% | 13.17% | 27.42% | 29.54% | +27.84% |
| Qwen3-32B | 48.60% | 32.40% | 19.37% | 33.46% | 35.60% | +33.52% |
| DeepSeek-V3.2 | 57.69% | 47.12% | 46.52% | 50.44% | 51.14% | +42.04% |
| DeepSeek-R1 | 62.50% | 53.90% | 54.26% | 56.89% | 57.40% | +38.47% |
| GPT-5.2 | 78.36% | 78.53% | **82.94%** | 79.94% | 79.54% | **+49.81%** |
| Gemini-3 Pro | 79.32% | 75.91% | 67.44% | 74.22% | 75.18% | +47.53% |
| Claude-Sonnet 4.6 | **83.65%** | **85.34%** | 80.62% | **83.20%** | **83.33%** | +41.85% |

## 4. Experiment

We organize our empirical analysis around two core research questions. RQ1 evaluates the performance disparity between ProFinAgent and standard baselines across hierarchical difficulty levels. This comparison validates the discriminative rigor of the ProFinR benchmark and confirms the efficacy of our agentic architecture. RQ2 investigates the structural mechanisms driving baseline failures. Through granular trajectory analysis, we attribute performance deficits to inherent architectural limitations rather than stochastic noise. This diagnosis explains why native models falter under the strict logical and temporal constraints of professional financial tasks.

### 4.1. Experiment Setup

We evaluate our framework on eight representative LLMs. These models include proprietary systems accessed through official APIs and open weight models deployed locally with SGLang on two NVIDIA A800 GPUs. For semantic retrieval, we use Qwen3-Embedding-0.6B (Zhang et al., 2025). Detailed experimental settings are provided in Section A.1.

### 4.2. Benchmark Evaluation

A granular analysis of Setting I (Native Inference) reveals a critical performance paradox inherent in state-of-the-art LLMs. As presented in Table 3, pre-trained models such as GPT-5.2 and Gemini-3 Pro exhibit a phenomenon we term "Generative Fluency" on complex L3 end-to-end inves-

tigation. For instance, GPT-5.2 achieves a deceptively high score of 65.89% on L3, such as report generation. However, this linguistic coherence masks a severe deficiency in factual grounding. When evaluated on foundational tasks, the same model's performance collapses, scoring only 16.82% on L1 retrieval and 19.37% on L2 calculation. We attribute this divergence to the models' reliance on "Recency Bias" and internal parametric memory. While large-scale pre-training allows them to construct semantically plausible market narratives, they lack the structural mechanism for real-time data verification and mathematical precision. As a result, their answers are often plausible but factually unsupported without the structural constraints of a structured workflow. This confirms that mere scaling of model parameters is insufficient for rigorous financial reasoning, as the absence of external verification tools inevitably leads to widespread data fabrication. Furthermore, we find that domain-specific training is not an alternative solution. Fin-R1 shows poor performance despite specialized financial training. Such models tend to overfit static historical datasets and fail under dynamic real-world scenarios. This rigid specialization suggests that training alone cannot produce reliable autonomous agents.

To illustrate the performance gap, we analyze execution trajectories for a representative L2 task: *"Check for MACD bearish divergence on the weekly chart for Salesforce (CRM) during Q3 2025."* As shown in Figure 3, the ReAct-based baseline exhibits severe sequential misalignment due to the absence of topological awareness. The

*Table 4.* Performance comparison on RestBench-TMDB and RestBench-Spotify. A higher score indicates better method performance. Bold figures highlight the maximum score per configuration. 'Pass' and 'Win' abbreviate pass rate and win rate.

| Model | Method | TMDB Pass | TMDB Win | AVG | Spotify Pass | Spotify Win | AVG |
|---|---|---|---|---|---|---|---|
| | Zero-shot | 33.28% | 50.00% | 41.64% | 26.44% | 50.00% | 38.22% |
| | CoT | 34.42% | 54.70% | 44.56% | 29.82% | 53.10% | 41.46% |
| | ReAct | 38.82% | 61.06% | 49.94% | 32.64% | 59.95% | 46.30% |
| GPT-4o-mini | DFSDT | 46.20% | 64.26% | 55.23% | 35.10% | 65.47% | 50.28% |
| | LATS | 51.33% | 66.67% | 59.00% | 39.81% | **72.85%** | 56.33% |
| | TOOLTREE | 55.17% | 70.40% | 62.79% | 42.08% | 72.18% | 57.74% |
| | **ProFinAgent** | **62.00%** | **77.00%** | **69.50%** | **52.63%** | 71.93% | **62.28%** |
| | Zero-shot | 56.28% | 50.00% | 53.14% | 49.54% | 50.00% | 49.77% |
| | CoT | 58.52% | 52.32% | 55.42% | 47.92% | 44.55% | 46.23% |
| | ReAct | 62.42% | 66.17% | 64.30% | 53.27% | 60.72% | 57.00% |
| GPT-4o | DFSDT | 66.57% | 69.08% | 67.82% | 55.48% | 71.63% | 63.55% |
| | LATS | 68.26% | 74.44% | 71.35% | 61.25% | 75.80% | 68.53% |
| | TOOLTREE | 72.40% | 75.59% | 74.50% | 60.87% | 78.84% | 71.36% |
| | **ProFinAgent** | **85.00%** | **77.50%** | **81.25%** | **68.42%** | **82.46%** | **75.44%** |

agent invokes *Talib_Technical_Indicators* before executing *fmp_historical_data*, violating the dependency between data acquisition and computation and triggering a "No Data" exception. Without structural constraints, the agent enters repeated trial-and-error behavior that often ends in fabricated outputs and loss of factual grounding. To explain the performance gap in our experiments, we attribute failures in native LLMs to three structural errors rather than random errors: Temporal Misalignment, Data Fabrication, and Tool Hallucination. Detailed in Table 10, these errors arise from the inherent limitations of unstructured inference. Lacking an explicit dependency graph, the reasoning process degenerates into token prediction, causing logical constraints to erode as execution length increases.

> **Analysis of Failure Case**
>
> ```
> Query: "Check for MACD bearish divergence on
> the weekly chart for Salesforce (CRM) during Q3
> 2025."
>
> Agent ToolChain:"task_details": [{"task_id":
> 1, "tool": "talib_technical_indicators",
> "status": "fail"}, {"task_id": 2, "tool":
> "fmp_historical_data", "status": "fail"}] }
>
> Analysis: "No available data for
> talib_technical_indicators."
> ```

*Figure 3.* Failure Case Analysis.

The results in Setting II confirm that ProFinAgent effectively converts "superficial fluency" into "grounded accuracy". Native LLMs often suffer from tool sequence misplacement and generative hallucinations. ProFinAgent dismantles this paradox by enforcing topological regularization upon the inference process. As shown in Figure 5, ProFinAgent retrieves optimal tools from the 13-category library while orchestrating a valid execution trajectory to preclude sequential mismatches. By integrating the Case-Based Memory with the DAG-based planner, we observe a significant im-

provement. GPT-5.2 exhibits a notable improvement in performance on foundational tasks. Its L1 and L2 scores rise significantly to 78.36% and 78.53%, respectively. This solid foundation propels the L3 success rate to a remarkable 82.94%. The framework bridges the gap between vague parametric memory and precise execution. The DAG explicitly encodes execution order. It ensures that calculation nodes activate only after prerequisite data nodes successfully populate the global context. This dependency-locking mechanism eliminates the temporal misalignment and data fabrication common in unstructured baselines.

In addition, ProFinAgent extends beyond improvements in reasoning accuracy to gains in computational efficiency. As shown in Figure 4, the topological independence induced by DAG design enables the parallel execution of non-dependent sub-tasks, alleviating the latency bottlenecks that commonly arise in complex agentic workflows. For DeepSeek-R1, the DAG-enabled pipeline decreases total execution time from 42,579.9 seconds to 22,547.2 seconds, resulting in a 47.1% reduction in latency. By effectively decoupling reasoning depth from inference time, our approach not only improves accessibility to complex reasoning but also demonstrates that structurally guided parallelism makes rigorous financial analysis.

### 4.3. ProFinAgent Evaluation

Beyond accuracy, the topological independence induced by the DAG design yields significant computational gains. Complex financial workflows often face severe latency bottlenecks. ProFinAgent resolves this by parallelizing non-dependent sub-tasks within the execution graph. The pipeline reduces total execution time for DeepSeek-R1 by 47.1%. This efficiency confirms that structurally guided parallelism makes rigorous financial analysis computationally feasible for real-world deployment. The framework effec-

*Table 5.* Ablation study on varying complexities (L1-L3). The **Baseline** uses ReAct. **w/ DAG** and **w/ CBM** denote the integration of the Directed Acyclic Graph and Case-Based Memory, respectively.

| Method | Qwen3-4B-Instruct-2507 | | | Qwen3-32B | | | DeepSeek-R1 | | |
|---|---|---|---|---|---|---|---|---|---|
| | L1 | L2 | L3 | L1 | L2 | L3 | L1 | L2 | L3 |
| **Baseline** | 4.32% | 0.00% | 0.00% | 5.28% | 0% | 0% | 17.78% | 13.08% | 29.45% |
| **w/ Tool Use** | 31.73% | 13.08% | 3.87% | 40.38% | 23.03% | 7.75% | 43.75% | 30.89% | 21.70% |
| **w/ DAG** | 37.50% | 17.28% | 3.10% | 44.71% | 25.13% | 10.85% | 51.44% | 42.40% | 33.33% |
| **w/ CBM** | 30.28% | 14.13% | 0.75% | 45.67% | 26.17% | 6.20% | 53.36% | 39.79% | 31.78% |
| **ProFinAgent** | **40.30%** | **28.79%** | **13.17%** | **48.60%** | **32.40%** | **19.37%** | **62.50%** | **53.90%** | **54.26%** |

tively decouples reasoning depth from inference latency.

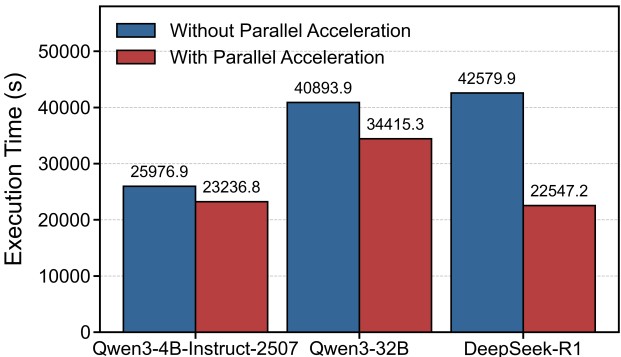

*Figure 4.* Analysis of DAG Parallelism on Agent Execution Time

### 4.4. Tool Planning Evaluation

We evaluate ProFinAgent on general benchmarks to verify domain-agnostic robustness. This assessment extends beyond financial reasoning to encompass broad tool-orchestration tasks. As shown in Table 4, ProFinAgent achieves state-of-the-art performance on both the RestBench TMDB and RestBench Spotify datasets. Our framework uses GPT-4o to achieve average accuracies of 81.25% on TMDB and 75.44% on Spotify, significantly outperforming advanced baselines like ToolTree (Yang et al., 2026). These results demonstrate that ProFinAgent generalizes effectively beyond financial environments to diverse non-financial tasks.

Tree-search frameworks such as DFSDT (Qin et al., 2023) and ToolTree explore large action spaces by sequentially scoring candidate actions with local heuristic rewards. This search strategy ignores global dependencies among tool calls, often selecting downstream actions before their prerequisites are met. Consequently, these methods may hallucinate tool arguments, such as fabricated movie identifiers used to query actor profiles before database retrieval completes. Such random exploration leads to structural errors and costly backtracking, exposing a fundamental lack of topological awareness. ProFinAgent resolves this issue by imposing deterministic topological constraints. It constructs an explicit dependency graph over tool calls, and its DAG planner

enforces global prerequisites before execution, blocking downstream nodes until upstream outputs are available. This design eliminates parameter hallucination, maintains structural consistency, and improves tool-orchestration efficiency.

Moreover, the CBM module enhances reasoning stability. Existing tool-planning frameworks treat each query in isolation, causing similar structural errors to recur across related tasks. This behavior reflects a form of execution amnesia. CBM addresses this problem through evolutionary adaptation that improves planning without parameter updates. Historical failures are encoded as negative constraints to prune invalid action regions, while validated trajectories provide high-quality positive templates that reduce exploration cost on new tasks. This bidirectional feedback guides the agent toward more reliable reasoning trajectories, turning superficial fluency into grounded accuracy.

### 4.5. Ablation Study

We analyze the individual contribution of the topological planner and the reflective memory module. As reported in Table 5, we compare the complete ProFinAgent with a ReAct-style baseline and with variants that enable only a single component.

**Finding 1: Providing naive LLMs with an extensive API library is insufficient for robust agentic reasoning.** The ReAct structure exposes a fundamental weakness when operating in a high-dimensional action space comprising 53 financial tools. Under this setting, DeepSeek-R1 attains only 21.70% accuracy on L3 tasks. Qualitative inspection of execution traces indicates that unstructured agents exhibit systematic sequential misalignment. In particular, they frequently invoke computational modules before the required data have been retrieved. This behavior demonstrates that the ReAct structure fails to use external tools systematically, instead degenerating into unstructured trial-and-error interactions.

**Finding 2: Integrating the dependency-aware DAG sub-**

### Analysis of Successful Case

```
Query: "Check for MACD bearish divergence on
the weekly chart for Salesforce (CRM) during Q3
2025."

Agent ToolChain:"task_details": [{"task_id":
1, "tool": "crawler_yfinance", "status":
"success"}, {"task_id": 2, "tool":
"talib_technical_indicators", "status":
"success"}] }
```

*Figure 5.* Successful Case Analysis.

**stantially reduces reasoning instability.** When compared with the ReAct baseline, the *w/ DAG* variant consistently improves performance. In particular, DeepSeek-R1's L3 accuracy increases to 33.33%. The DAG serves as a logical guardrail by explicitly resolving information dependencies before execution. This design prevents the temporal hallucinations observed in the baseline. The resulting structural constraint ensures that agent behavior follows a valid causal sequence rather than unconstrained generation.

**Finding 3: Reflective memory improves execution precision through dual feedback.** Case-Based Memory (CBM) provides clear benefits for execution precision. The *w/ CBM* variant consistently outperforms the ReAct baseline on retrieval-intensive L1 tasks. For example, DeepSeek-R1 achieves an accuracy of 53.36%, compared with 43.75% under the baseline. This improvement arises from the structured reuse of archived execution experiences. Successfully validated trajectories are stored as successful cases. These cases function as reusable templates for tool choosing. By retrieving such high-fidelity analogues, the agent anchors its current planning process in previously verified execution paths. This analogy reduces structural uncertainty and promotes the proper use of tools. At the same time, failed cases contribute complementary supervisory signals. Historical errors are encoded as negative constraints within the memory module. During planning, these constraints are used to prune invalid regions of the action space. In particular, the system explicitly marks erroneous API combinations and ineffective tool sequences. This bidirectional feedback mechanism suppresses the recurrence of domain-specific failure modes while systematically reinforcing effective execution strategies.

**Finding 4: The ProFinAgent framework achieves a qualitative leap through module integration.** It surpasses both single-module variants by a wide margin. DeepSeek-R1 reaches a dominant 54.26% on L3 tasks. This score significantly exceeds both the DAG-only (33.33%) and CBM-only (31.78%) configurations. The results validate our core hypothesis. Structural planning provides the necessary framework for reasoning. The CBM enables our framework to couple dependency-consistent reasoning with adaptive robustness, improving stability without sacrificing flexibility.

## 5. Related Work

### 5.1. Financial Benchmarks

Early benchmarks such as FinQA (Chen et al., 2021) and TAT-QA (Zhu et al., 2021) primarily focus on static numerical extraction from textual and tabular sources. Subsequent benchmarks, such as FinBen (Xie et al., 2024), expand coverage to general financial knowledge but do not model interactions with dynamic environments. FinSearchComp (Hu et al., 2025) adds tool-augmented retrieval but is limited to L1 single-hop verification. FinDeepResearch (Zhu et al., 2025) evaluates advanced agents across rigorous analytical tasks. As a result, existing benchmarks fall short of capturing the complexity of real-world financial workflows, particularly multi-step quantitative reasoning (L2) and autonomous report generation (L3). Moreover, most prior evaluations emphasize final answer accuracy while overlooking the logical soundness of intermediate execution trajectories.

### 5.2. Financial Agent

Recent work in financial agents has shifted toward autonomous multi-agent systems. QuantAgent (Xiong et al., 2025) and FINCON (Yu et al., 2024) decompose market analysis into specialized agents for technical, fundamental, and macroeconomic reasoning, improving robustness through modular coordination. FinThink (Zeyu et al.) introduces collaborative frameworks to enhance financial reasoning. To address market non-stationarity, TradingGPT (Li et al., 2023) and HedgeAgents (Li et al., 2025) adopt hierarchical memory architectures that separate short-term working memory from long-term episodic memory. In addition, SocioDojo (Cheng & Chin, 2024) extends agent learning to open multimodal environments, while FinPersona (Takayanagi et al., 2025) explores persona-based adaptation for financial advising. Besides, Fin-R1 (Liu et al., 2025) applies reinforcement learning to improve reasoning trajectories. FinCoT (Nitarach et al., 2025) establishes a structured thinking process by using expert demonstrations. However, most systems rely on simple reasoning structures (Yao et al., 2022) (Yao et al., 2023), which makes long-term financial reasoning prone to error accumulation and illusions.

## 6. Conclusion

In this paper, we introduce ProFinR, a financial reasoning benchmark that measures agents' abilities across the three levels of financial reasoning tasks. We also built the Financial Tool Universe, a library of 53 finance tools. We further propose ProFinAgent, a structure-augmented framework that imposes structured execution on financial reasoning. Future work will explore multi-agent market simulation and real-time risk management.

## Acknowledgements

This work was supported by the National Key Research and Development Program of China under 2024YFC3307603. This work was supported in part by Tsinghua University Toyota Research Center. This work was also supported by the Zhongguancun Academy, (Grant No. C20250201).

## Impact Statement

This work introduces ProFinR and ProFinAgent to advance the evaluation and reliability of tool-augmented financial reasoning, helping reduce hallucinations and improve verifiable analysis in high-stakes settings. Potential negative impacts include misuse for automating or scaling misleading financial advice or trading; we position our contributions primarily as research infrastructure and recommend responsible deployment with human oversight.

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

# A. Experimental Setup and Reproducibility

## A.1. Foundation Models and Deployment Infrastructure

To conduct a comprehensive evaluation of our proposed framework, we selected eight representative Large Language Models (LLMs) spanning different scales and access paradigms, as detailed in Table 6. For proprietary models, we used their official APIs to benchmark performance against industry-leading standards. To ensure rigorous control over the inference parameters and latency measurements, all open-weight models were deployed locally. To enhance our workflow, we standardized the semantic embedding process to minimize variability. We employed Qwen3-Embedding-0.6B (Zhang et al., 2025) as the underlying embedding model. We employed the high-throughput SGLang framework on dual NVIDIA A800 GPUs to ensure sufficient VRAM for high-precision and long-context inference.

*Table 6.* **Model Specifications (Open Weights vs. API).**

| Model Name | Size | Form | Access |
|---|---|---|---|
| DeepSeek-R1 (DeepSeek AI, 2025a) | – | API | Remote |
| DeepSeek-V3.2 (DeepSeek AI, 2025b) | – | API | Remote |
| GPT-5.2 (OpenAI, 2025) | – | API | Remote |
| Gemini-3 Pro (Google DeepMind, 2025) | – | API | Remote |
| Claude-Sonnet 4.6 (Anthropic, 2026) | – | API | Remote |
| Qwen3-4B-Instruct-2507 (Team, 2025) | 4B | Open Weights | Local |
| Fin-R1 (Liu et al., 2025) | 7B | Open Weights | Local |
| Qwen3-32B (Team, 2025) | 32B | Open Weights | Local |

## A.2. Baseline Tool Planning Frameworks

We compare our method against several prominent tool planning architectures. These baselines represent diverse strategies for autonomous reasoning and execution.

Zero-shot: The model generates answers directly. It skips all intermediate reasoning.

Chain-of-Thought (CoT) (Wei et al., 2022): Models produce explicit logical steps before producing the final response.

ReAct (Yao et al., 2022): This framework combines reasoning with actions. Agents generate internal thoughts. They then execute corresponding tool calls.

DFSDT (Qin et al., 2023): This method employs Depth First Search strategies. It explores potential trajectories in the action space. Backtracking logic refines the planning path.

LATS (Language Agent Tree Search) (Zhou et al., 2023): This approach integrates tree search with external feedback. Systems evaluate diverse reasoning branches. Reflection cycles enhance decision quality.

ToolTree (Yang et al., 2026): This paradigm leverages Monte Carlo Tree Search. It incorporates a dual-stage evaluation process. The architecture executes bidirectional pruning. Such designs maximize search efficiency for complex tasks.

## A.3. Prompting Frameworks

**Workflow - Test Agent Profile**

```
    You are a Financial Full-Pipeline AI Agent designed to solve end-to-end
financial tasks using structured reasoning, DAG planning, and tool execution
via the Model Context Protocol (MCP).
```

**Workflow - Judge Agent Profile**

```
    You are an expert judge for AI agents.  Task:  Evaluate if the agent
correctly solved the financial query.
```

### Workflow - Choose Tool Types

```
{context}

{query} is the context of the current task.

{tool_types} are the types of tools that you can use to solve the current
task.

You need to choose the types of tools that you need to use to solve the
current task.

The output format is as follows (must be an exact match, do not add any
parameters):

{{ "tool_types": ["tool_type1", "tool_type2", "tool_type3"] }}
```

### Workflow - ToolChain

```
{context}

{query} is the context of the current task.

{tools} is the tools that you can use to solve the question.

{notebook} is the notebook that is most similar to this task, you can
refer to the experience in the notebook to avoid potential problems.

The entity names and timestamps in the toolchain need to be confirmed to
be consistent with the data in the problem.

You must output JSON only, strictly following this schema (do NOT add
extra top-level keys):

{{ "toolchain_calls": [ {{ "tool": "tool_name", "arguments": {{
"argument1": "value1", "argument2": "value2" }} ] }} }}
```

### Workflow - Tool Dependencies

```
{context}

{result} is the tool planning result of the current task.

Analyze the toolchain calls and determine the logical dependencies
between tasks.

IMPORTANT RULES: 1. You MUST preserve all task IDs, tool names, and
arguments exactly as they appear in the input. 2. You MUST output ALL
tasks from the input, with the same IDs, tool names, and arguments. 3.
You ONLY need to add the "dependencies" field to each task based on logical
dependencies.
```

```
   Dependency Analysis:  - If a task's arguments reference the output of
another task (e.g., file paths, data from previous steps, output from other
tools), add that task's id to the dependencies list.  - If a task needs to
wait for another task to complete before it can start, add that task's id to
the dependencies list.  - If tasks are completely independent and can run
in parallel, the dependencies list is empty [].  - A task cannot depend on
itself (no circular dependencies).

   You need to output the tool dependencies of the current task based on the
toolchain calls.

   The output format is as follows (must be an exact match, do not add any
parameters):

   [ {{ "id":  1, "tool_name":  "...", "arguments":  {{ "argument1":
"value1", "argument2":  "value2" ...  }}, "dependencies":  [] }} ]
```

**Workflow - Final Answer**

```
   {context}

   The current task is:  {task}.

   {result} is valid information obtained through the toolchain.

   {notebook} is the notebook that is most similar to this task, you can
refer to the experience in the notebook to avoid potential problems.

   You need to provide a specific answer based on the current problem and
the results output by the toolchain, such as Revenue CAGR: 14.5

   You can only answer based on your own knowledge and the data within the
required time frame.  Do not use any information outside the frame of the
query time needed or fabricated content.

   The output format is as follows (must be an exact match, do not add any
parameters):

   {{ "final_answer":  "..." }}
```

**Workflow - Self Reflection**

```
   {context}

   The current task is:  {task}.

   The toolchain calls are:  {result}.

   The dag results are:  {dag_results}.
```

```
    The types of tools needed for the reference answer include:
{reference_tools}.

    The final answer is: {final_answer}.

    The Reference answer for this task is: {reference_answer}.

    You need to output the self-reflection based on the current task,
toolchain result, dag results, final answer, and reference answer.

    The output format is as follows (must be an exact match, do not add any
parameters):

    {{ "task": "...", "dag_results": "...", "self_reflection": "..." }}
```

### Workflow - No Tool Answer

```
    {context}

    The current task is: {task}.

    You are a React-style financial answering agent for financial reasoning
tasks. Before producing the final answer, reason internally through: 1.
Understand the user's exact question, required entities, metrics, and
time range. 2. Identify what information is already available in the
task context. 3. Verify that the answer can be grounded in the given
context and generally available knowledge within the required time frame.
4. Reject any fabricated numbers, unsupported facts, or information outside
the required time frame. 5. Compose a concise, direct final answer that
addresses the task.

    Do not output your internal reasoning.

    You must output JSON only, strictly following this schema (do NOT add
extra top-level keys):

    {{ "final_answer": "..." }}
```

### Workflow - Judge

```
    {context}

    User Request: {query} is the financial query.

    Agent Execution: {agent_execution} is the agent's execution output.

    The true answer is {true_answer}.

    The agent's output determines whether the question has been answered
correctly. The entity names only need to indicate that they are the same,
```

```
not necessarily identical.  The price can fluctuate by about 1%.  The final
report then analyzes whether the answer meets the requirements.

   Mark the result as PASS or FAIL. Do not add any other text.

   The output format is as follows in JSON format (must be an exact match,
do not add any parameters):  {{ "result":  "PASS" or "FAIL" }}
```

**Workflow - Answer Straightforward Tool**

```
   {context}

   The current task is:  {task}.

   {tools} are the tools that you can use to solve the question.

   The entity names and timestamps in the toolchain need to be confirmed to
be consistent with the data in the problem.

   You must output JSON only, strictly following this schema (do NOT add
extra top-level keys):  {{ "toolchain_calls":  [ {{ "tool":  "tool_name",
"arguments":  {{ "argument1":  "value1", "argument2":  "value2" }} }} ] }}
```

## B. Benchmark Construction and Evaluation Methodology

### B.1. Data Construction and Quality Assurance

To ensure the reliability of ProFinR, particularly for complex L3 End-to-End Investigation tasks, we implemented a rigorous three-stage data construction pipeline involving domain experts.

**1. Data Provenance and Diversity.** We sourced raw financial data from authoritative repositories, including SEC EDGAR (10-K/10-Q filings), earnings call transcripts, and macro-economic indicators from 2020 to 2025. To prevent data contamination, we strictly excluded data focused on private/long-tail data. The dataset covers 7 sectors to ensure broad domain coverage.

**2. Expert-Driven Annotation Protocol.** Our annotation team comprised 15 domain experts, including Finance PhD candidates and CFA (Chartered Financial Analyst) Level II+ holders, unlike generic benchmarks.

- *L1/L2 Annotation:* Experts formulated queries based on specific financial statements and manually verified the calculation paths.

- *L3 Ground Truth Construction:* For open-ended L3 tasks (e.g., "Generate a Q3 investment report for AAPL"), defining a single "correct" text is infeasible. Instead, we adopted a **Key Information Point (KIP)** strategy. Experts defined a "Gold Checklist" for each task, containing essential numerical facts (e.g., "Revenue=$89.5B"), required risk factors (e.g., "China supply chain"), and the logical deduction chain. A model's output is deemed correct only if it covers these KIPs and derives a consistent conclusion.

**3. Quality Control.** We enforced a *Double-Blind Cross-Verification* process. Each task was evaluated and annotated separately by two experts. In cases of discrepancy, a Senior Financial Analyst acted as the arbitrator.

The final score $S_{\text{total}}$ employs a complexity-adaptive normalization scheme to ensure scale invariance across task hierarchies ($S_{\text{total}} \in [0, 1]$). We introduce a dynamic scaling factor $Z$ that normalizes the weighted sum of active evaluation dimensions. The metric is formalized as:

$$S_{\text{total}}(y, y^*) = \frac{1}{Z}\left(\alpha S_{\text{val}} + \beta S_{\text{tool}} + \gamma \mathbb{I}_{\text{L3}} S_{\text{sound}}\right) \tag{8}$$

where $\mathbb{I}_{\text{L3}}$ is the indicator function for Level 3 tasks. The normalization coefficient $Z$ represents the sum of active weights:

$$Z = \alpha + \beta + \gamma \mathbb{I}_{\text{L3}} \tag{9}$$

Given the weight configuration ($\alpha = 0.2, \beta = 0.3, \gamma = 0.5$), this scheme automatically adapts to task complexity. For L1 and L2 tasks ($\mathbb{I}_{\text{L3}} = 0$), the soundness term vanishes and $Z$ adjusts to 0.5, effectively rescaling the foundational metrics to the full unit interval. For L3 tasks ($\mathbb{I}_{\text{L3}} = 1$), the full spectrum of metrics applies with $Z = 1.0$. A task is considered successfully resolved if and only if $S_{\text{total}} > 0.6$.

### B.2. Taxonomy of the Financial Tool Universe

*Table 7.* Taxonomy and Functional Descriptions of the Financial Agent Toolset.

| Tool Category | Specific Content & Capabilities |
|---|---|
| Market Data | Retrieves raw market data (OHLCV, trading calendars) for stocks and other assets. |
| Corporate Fundamentals | Accesses financial statements, financial ratios, valuation metrics, risk indicators, ESG scores, etc. |
| Macroeconomic Data | Fetches macroeconomic time series such as GDP, inflation rates, interest rates, and treasury yields. |
| News & Sentiment | Retrieves news articles, earnings call transcripts, and social media text streams. |
| Regulatory Filings | Accesses and structures SEC filings (e.g., 10-K, 10-Q, 13F). |
| Web Scraping | Crawls targeted webpages and extracts structured textual content. |
| Data Processing | Performs local data cleaning, preprocessing, and format conversion. |
| Indicator & Factor Calculation | Computes technical indicators, Factor IC/IR, backtesting metrics, and text sentiment scores. |
| Model Training | Trains quantitative or factor models (including Tree models, Deep Learning, and RL). |
| Search & Knowledge | Executes general search queries, encyclopedia lookups, and model list retrieval. |
| Time-Series Forecasting | Uses pre-trained models to perform time-series predictions. |
| Alternative Market Data | Provides market data for cryptocurrencies, foreign exchange (Forex), and commodities. |
| Report Generation | Synthesizes PDF reports combining CSV/JSON data, visualizations, and LLM-driven analysis. |

## C. Qualitative Analysis and Failure Diagnostics

### C.1. Structural Weaknesses of Traditional Prompting

*Table 8.* Performance Comparison of ProFinAgent and Prompting Baselines Across Different Foundation Models

| Model | Method | L1 | L2 | L3 |
|---|---|---|---|---|
| | ReAct | 69.23% | 38.74% | 46.51% |
| GPT-5.2 | Tree-of-Thoughts | 66.35% | 47.64% | 66.66% |
| | **ProFinAgent** | 78.36% | 78.53% | **82.94%** |
| | ReAct | 73.07% | 46.60% | 45.74% |
| Gemini-3 Pro | Tree-of-Thoughts | 78.36% | 55.50% | 35.65% |
| | **ProFinAgent** | 79.32% | 75.91% | 67.44% |
| | ReAct | 65.87% | 61.78% | 65.89% |
| Claude-Sonnet 4.6 | Tree-of-Thoughts | 68.75% | 69.63% | 63.56% |
| | **ProFinAgent** | **83.65%** | **85.34%** | 80.62% |

We further evaluate advanced language models under traditional prompting frameworks. Table 8 reports the results across different prompting architectures. ReAct serves as the main reactive baseline, and ToT serves as a search-based baseline for multi-step planning. Overall, ToT performs better than ReAct in most settings. It explores multiple reasoning paths and

*Table 9.* Representative failure modes of trained models on ProFinR. These examples show that static training does not equip an agent with dynamic retrieval and structured execution.

| Error Category | Query | Observed Answer |
|---|---|---|
| **1. Inability to Compute Technical Indicators** (Fails to calculate due to lack of market data retrieval tools) | Calculate the RSI and MACD indicators for NVIDIA (NVDA) for the first half of 2024 to identify potential overbought conditions. | The RSI and MACD indicators for NVIDIA (NVDA) for the first half of 2024 cannot be calculated without real-time or historical price data access, which is not available in this environment. Therefore, a precise numerical result cannot be provided. |
| **2. Factual Hallucination** (Guesses incorrect facts due to reliance on internal memory without search tools) | Who is the current CFO of Amazon? | Andy Jassy |
| **3. Lack of Real-Time Market Data Retrieval** (Fails to fetch historical or live pricing data) | I need to review Apple's (AAPL) price movements during 2023 to prepare for a volatility analysis. | I cannot provide specific price movement data for Apple (AAPL) during 2023, as no real-time or historical price data has been accessed or retrieved in this context. Please consult a financial data source such as Yahoo Finance, Bloomberg, or Google Finance for accurate historical price movements. |
| **4. Inability to Access Regulatory Filings** (Fails to retrieve the required 10-K or 10-Q documents for fundamental analysis) | Get the full text of the Management's Discussion and Analysis (MD&A) section from the 2023 Form 10-K of Netflix (NFLX). | The full text of the Management's Discussion and Analysis (MD&A) section from the 2023 Form 10-K of Netflix (NFLX) is not available in this environment. Please refer to the official SEC EDGAR database or Netflix's investor relations website for the complete document. |
| **5. Lack of External Search for Sentiment Analysis** (Fails to fetch news or social media data for qualitative NLP tasks) | Monitor the latest news regarding GameStop and analyze the sentiment to generate a summary report for the trading desk. | I cannot provide a final answer as I do not have real-time access to current news or sentiment analysis data for GameStop. Please consult a financial news source or sentiment analysis tool for the latest information. |
| **6. Inability to Access Macroeconomic Databases** (Fails to retrieve FRED, CPI, or GDP metrics) | Download the monthly US Consumer Price Index (CPI) data for the entire year 2023. | The monthly US Consumer Price Index (CPI) data for the entire year 2023 is not available in this context. Please consult official sources such as the U.S. Bureau of Labor Statistics (BLS) for accurate and up-to-date data. |
| **7. Temporal Limitation and Forecasting Failure** (Fails to process queries involving future/recent dates beyond its training cutoff) | Closing price of spot Crude Oil WTI (CL=F) on September 30, 2024. | Data for the closing price of spot Crude Oil WTI (CL=F) on September 30, 2024, is not available as of now. The date is in the future, and real-time or future market data cannot be retrieved. |

compares alternative solutions. In contrast, ReAct follows a single trajectory and makes greedy decisions at each step. This design is sensitive to early errors. When a mistake occurs, it often spreads throughout the reasoning process. ToT addresses this issue by exploring branches and evaluating intermediate results. Consequently, it is better equipped to recover from suboptimal decisions.

ProFinAgent addresses the structural weakness of pure prompting methods. ReAct and ToT often overlook strict dependencies between tools, which often lead to invalid execution in reasoning workflows. ProFinAgent uses a DAG planner to enforce topological order. The CBM module further improves reliability through long-term memory. Existing baselines do not learn effectively from past failures or successes. In contrast, ProFinAgent uses archived cases to prune erroneous paths and support better decisions.

### C.2. Limitations of Domain-Specific Training

We further analyze why trained models continue to fail on ProFinR. Our purpose is to identify the primary failure modes and explain why domain-specific training alone is insufficient for ProFinR. During evaluation, ProFinR not only tests financial knowledge but also tests whether an agent can retrieve external evidence, choose valid tools, and follow a correct execution order. This requirement makes the benchmark fundamentally different from static question answering.

Table 9 summarizes the major error categories observed in the trained models. These failures reveal a common gap between

offline training and dynamic tool use. In practice, trained models often fail in three critical ways, as shown in Table 10. First, these models rely too heavily on parametric memory and often generate plausible answers that are not grounded in external evidence. This weakness is especially severe on ProFinR, where many questions require current facts, historical records, or direct document retrieval. Second, they struggle to complete multi-step workflows that depend on strict tool ordering. Even when the required tools are available, the model may skip retrieval, call tools with invalid inputs, or generate conclusions before the necessary evidence is collected. Third, they show limited robustness when the task requires interaction with real financial systems rather than static question answering. As a result, domain-specific training improves fluency and terminology, but it does not ensure faithful retrieval, valid execution, and grounded reasoning. This is the main reason why trained models still perform poorly on ProFinR.

This analysis also explains why Fin-R1 still performs poorly on ProFinR after financial training. We add Fin-R1 because it is one of the few advanced financial agents. However, the results show that its performance remains limited on ProFinR. During evaluation, we find that Fin-R1 merely generates basic report outlines or plain text for tool execution but completely fails to solve actual problems. While it could generate fluent financial text, such fluency fails to address the core challenges inherent in ProFinR. The benchmark requires real data access, reliable tool selection, and strict execution order. These abilities are not guaranteed by domain-specific training on static corpora. Therefore, training on financial data alone does not produce a reliable financial agent.

## C.3. Taxonomy of Hallucinations in Financial Agents

*Table 10.* Taxonomy of Hallucinations in Financial Agents with Representative Case Studies.

| Category | Query & Agent Answer | Analysis |
|---|---|---|
| **Temporal Misalignment** | *Q: What were the dividend payout dates and amounts for Coca-Cola (KO) throughout the calendar year 2023?* *A: The dividend payout dates for 2023 were: February 17, 2023 ($0.44 per share); May 19, 2023 ($0.44 per share).* | *The agent incorrectly reused 2022 dividend information and reported it as 2023 data.* |
| **Data Fabrication** | *Q: Get the Free Cash Flow (FCF) data for Intel (INTC) for each quarter of 2023.* *A:* `{ "2023-Q1": "-7.48 billion", "2023-Q2": "-3.00 billion", ...}` | *The numerical values are entirely fabricated and are not supported by Intel's financial disclosures.* |
| **Tool Hallucination** | *Q: I need to review Apple's (AAPL) price movements during 2023 to prepare for a volatility analysis.* *A:* `{ "function": "get_historical_data", "parameters": { "symbol": "AAPL", ...} }` | *The agent invokes a non-existent function that is not available in the execution environment.* |

---

**Algorithm 1** ProFinAgent: Evolutionary Financial Reasoning

---

1: **Require:** User Query $q$, Tool Set $\mathcal{A}$, Financial Database $\mathcal{D}$, Memory $\mathcal{M}$, LLM $\Pi_\theta$
2: **Require: Ground Truth** $y^*$ *// Available in training/optimization phase*
3: **Ensure:** Final Answer $y$, Updated Memory $\mathcal{M}'$
4: *// Phase 1: Perception & Retrieval*
5: $m_{\text{hist}} \leftarrow \text{KNN}(q, \mathcal{M}, k)$
6: $P_{\text{ctx}} \leftarrow \text{PROMPT}(q, m_{\text{hist}})$
7: $\mathcal{A}_{\text{sub}} \leftarrow \text{FILTERTYPE}(q, \mathcal{A})$
8: $\mathcal{A}^* \leftarrow \text{RANKSEMANTICS}(q, \mathcal{A}_{\text{sub}})$
9: *// Phase 2: Structural Planning*
10: $G(V, E) \leftarrow \Pi_\theta(P_{\text{ctx}}, \mathcal{A}^*)$ **subject to** DAG constraints
11: $\mathbb{L} = \{L_1, \ldots, L_K\} \leftarrow \text{LAYEREDTOPSORT}(G)$
12: $C \leftarrow \emptyset$
13: *// Phase 3: Dependency-Aware Parallel Execution*
14: **for** $k = 1$ **to** $K$ **do**
15:     $O_k \leftarrow \emptyset$
16:     **for all** node $v_i \in L_k$ **in parallel do**
17:         $p_i \leftarrow \text{ResolveInputs}(C, \{(u, v_i) \in E\})$
18:         $o_{\text{raw}} \leftarrow \text{EXECUTE}(v_i, p_i; \mathcal{D})$
19:         **if** $\text{Length}(o_{\text{raw}}) > \tau_{\text{len}}$ **then**
20:             $o_i \leftarrow \text{PARDISTILL}(o_{\text{raw}}, q)$
21:         **else**
22:             $o_i \leftarrow o_{\text{raw}}$
23:         **end if**
24:         $O_k \leftarrow O_k \cup \{o_i\}$
25:     **end for**
26:     $C \leftarrow C \cup O_k$
27: **end for**
28: *// Phase 4: Synthesis*
29: $y_{\text{rep}} \leftarrow \text{RetrieveReport}(C)$
30: $y \leftarrow \Pi_\theta(P_{\text{ctx}} \oplus C \oplus y_{\text{rep}})$
31: *// Phase 5: Supervised Evolution*
32: *// Evaluate prediction $y$ against ground truth $y^*$*
33: $s \leftarrow \text{JUDGE}(y, y^*)$
34: *// Generate reflection on the gap between $y$ and $y^*$*
35: $f_{\text{new}} \leftarrow \text{SELFREFLECT}(q, G, y, s, y^*)$
36: **if** $s < \tau_{\text{thres}}$ **then**
37:     *// Archive failure case with ground-truth correction*
38:     $\mathcal{M}' \leftarrow \mathcal{M} \cup \{(q, f_{\text{new}}, y^*)\}$
39: **else**
40:     *// Archive success case using the model's own output*
41:     $\mathcal{M}' \leftarrow \mathcal{M} \cup \{(q, f_{\text{new}}, y)\}$
42: **end if**
43: **return** $y, \mathcal{M}'$

---

