# OpenReview forum: "Towards Professional-Grade Financial Agents: Benchmarking, Tooling, and Structured Reasoning"
_ICML.cc/2026/Conference — ICML 2026 regular_

### Official Review · Reviewer_tvYj · 2026-03-02

**Soundness:** 2
**Presentation:** 3
**Significance:** 2
**Originality:** 3
**Overall Recommendation:** 3
**Confidence:** 4

**Summary:**

The paper introduces, ProFinR, a benchmark for evaluating the performance of LLMs in financial analysis. It combines data retrieval, data analysis, and end-to-end tasks, as well as tool use, across four different domains of finance. In addition, the paper develops, ProFinAgent, a structured framework for utilizing LLMs in conducting financial analysis -- and evaluates the performance of different LLMs within that framework, including against (baseline) LLMs without that framework.

**Compliance With Llm Reviewing Policy:**

Affirmed.

**Key Questions For Authors:**

* Are all 528 tasks available in the anonymized repo? If not, could you make these available (at least to ICML reviewers)?
* Why is the comparison between ProFinAgent and Baseline (as in Table 3) only evaluated on ProFinR, and not on other (existing) finance benchmarks?
* Regarding the LLMs tested: 1. Which specific GPT-5 is evaluated? 2. Why does the evaluation use such an old DeepSeek Model (R1) ? 3. Why is Claude not among the LLMs evaluated?

**Limitations:**

See weaknesses above.

**Strengths And Weaknesses:**

Strengths
* The paper aims to fill an important gap in existing evaluation of LLMs on financial analysis, as well-illustrated in table 1 (top of p. 2).
* The authors provided access to the anonymized code repo, which is very helpful.
* Parts A.1 and A.2 of the appendix are relatively clear, and provide basic and much-needed information regarding the creation of the ProFinR benchmark.
* The annotation team was comprised of domain experts, and the specific roles they played is explained in some detail.
* The divergence for some LLMs between L1 and L2 scores vs L3 scores -- which the authors call “generative fluency” -- is an interesting and important finding.
* The paper is generally well-written.

Weaknesses
* Beyond the examples of tasks provided in Table 2, I couldn’t seem to find additional tasks comprising the benchmark (of 528 tasks). My apologies if I missed this, and it’s included already.
* Despite engaging domain experts from finance (as per A.2), the paper does not explain what underlying construct(s) the ProFinR benchmark actually seeks to measure -- as emphasized in, for example, https://arxiv.org/abs/2505.10573 and https://arxiv.org/abs/2411.12990. For example, the abstract refers to “realistic financial workflows” but there is no indication or explanation in the paper regarding the extent to which (if any) ProFinR benchmark captures the real-world work of finance professionals, or whether it is even meant to. The fact that the tasks are “expert-designed” does not alone address this issue.
* The ostensible advantages of the ProFinAgent framework relative to the baseline (naïve) use of LLMs (as per Table 3) is only established through evaluations on the ProFinR benchmark. Given there are many existing financial analysis benchmarks for LLMs (as the authors discuss in detail), why were none of these tested? Surely, if the ProFinAgent is superior to naïve (framework-less) use of LLMs, then that difference in performance would also show up on other finance benchmarks.
* There are also issues concerning the particular LLMs that were test (see key questions below).

---

> ### Author Rebuttal · Authors · 2026-03-31
>
> We sincerely thank the reviewer for the positive assessment and the thoughtful summary.  We have carefully addressed your questions below.
>
> **W1 & Q1 (Availability of All 528 Tasks).**
>
> We are very happy to share our dataset and have now uploaded all 528 task queries to our anonymized repository. To strictly protect the validity of our benchmark and prevent any potential data contamination, we will temporarily not release the specific reference answers during this review process. Once the paper is successfully accepted, we will publish the complete dataset, including all ground truth answers, on Hugging Face.
>
> **W2 (The Underlying Constructs of ProFinR).**
>
> We completely agree with the observation regarding the need to clearly define the underlying constructs of our benchmark. In contrast to general benchmarks, ProFinR establishes strong criterion validity by explicitly measuring a model's capacity to execute end-to-end financial workflows. To ensure content validity, our 528 tasks were not merely crowdsourced but were rigorously designed by 15 domain experts, including Finance PhD candidates and CFA charterholders who mapped real-world analyst activities into our evaluation framework. To address the socio-technical gap and ensure external validity, we move beyond static text evaluations by forcing models to interact with a dynamic Financial Tool Universe comprising 53 real-world APIs, which accurately simulates the high-velocity data streams professionals face daily. Ultimately, our benchmark directly operationalizes the criteria of professional financial analysis, meaning our tasks do not just test abstract reasoning but specifically measure whether a model can reliably retrieve, compute, and synthesize financial data to produce accurate investment reports exactly as required in the real world.
>
> **W3 & Q2 (Additional experiments on other datasets).**
>
> We fully agree with the observation regarding the limitations of existing datasets. As discussed in our manuscript, existing financial benchmarks primarily focus on static document question answering or basic sentiment analysis. These benchmarks evaluate single-shot numerical reasoning over frozen text and do not support dynamic external tool execution or complex workflow planning. Therefore, they are fundamentally incompatible with testing our framework. To rigorously validate our tool planning capabilities, we evaluated ProFinAgent on RestBench to compare it with advanced planning approaches such as the recently published ICLR 2026 TOOLTREE algorithm (https://arxiv.org/abs/2603.12740). The results prove our architecture possesses advanced tool orchestration capabilities that successfully overcome the limitations of static datasets.
>
> | Model | Method | TMDB Pass | TMDB Win | TMDB AVG | Spotify Pass | Spotify Win | Spotify AVG |
> | :--- | :--- | :--- | :--- | :--- | :--- | :--- | :--- |
> | **GPT-4o-mini** | Zero-shot | 33.28% | 50.00% | 41.64% | 26.44% | 50.00% | 38.22% |
> | | CoT | 34.42% | 54.70% | 44.56% | 29.82% | 53.10% | 41.46% |
> | | ReAct | 38.82% | 61.06% | 49.94% | 32.64% | 59.95% | 46.30% |
> | | DFSDT | 46.20% | 64.26% | 55.23% | 35.10% | 65.47% | 50.28% |
> | | LATS | 51.33% | 66.67% | 59.00% | 39.81% | **72.85%** | 56.33% |
> | | TOOLTREE | 55.17% | 70.40% | 62.79% | 42.08% | 72.18% | 57.74% |
> | | **ProFinAgent** | **62.00%** | **77.00%** | **69.50%** | **52.63%** | 71.93% | **62.28%** |
> | **GPT-4o** | Zero-shot | 56.28% | 50.00% | 53.14% | 49.54% | 50.00% | 49.77% |
> | | CoT | 58.52% | 52.32% | 55.42% | 47.92% | 44.55% | 46.23% |
> | | ReAct | 62.42% | 66.17% | 64.30% | 53.27% | 60.72% | 57.00% |
> | | DFSDT | 66.57% | 69.08% | 67.82% | 55.48% | 71.63% | 63.55% |
> | | LATS | 68.26% | 74.44% | 71.35% | 61.25% | 75.80% | 68.53% |
> | | TOOLTREE | 72.40% | 75.59% | 74.50% | 60.87% | 78.84% | 71.36% |
> | | **ProFinAgent** | **85.00%** | **77.50%** | **81.25%** | **68.42%** | **82.46%** | **75.44%** |
>
> **W4 & Q3 (Clarification on LLM Selection and Specification).**
>
> We sincerely thank the reviewer for pointing out the need for clearer model specifications and apologize for the ambiguity in our original manuscript. We want to clarify that the GPT-5 model evaluated in our study is specifically the latest GPT-5.2 version. Regarding our choice of the DeepSeek R1 model, we intentionally included it to comprehensively observe how different models perform on our benchmark and to analyze how various structured reasoning methods impact these diverse models. Furthermore, we have conducted additional comprehensive experiments using the Claude 4.6 model. The results clearly show that our ProFinAgent framework achieves exceptional performance with Claude 4.6.
>
> | Model | Method | L1 | L2 | L3 |
> | :--- | :--- | :--- | :--- | :--- |
> | **Claude-4.6** | Baseline | 38.94%| 32.98%| 54.26%|
> | |ReAct | 65.87%| 61.78%| 65.89%|
> | | Tree-of-Thoughts | 68.75%| 69.63%| 63.56%|
> | | **ProFinAgent** | **83.65%** | **85.34%** | **80.62%** |

---

> > ### Author Rebuttal · Reviewer_tvYj · 2026-04-03
> >
> > The explanation of the underlying concept being measured was inadequate, and did not offer meaningful clarity. Also, I would want to see the results on additional finance benchmarks, not just one more benchmark. Finally, I'm still uncertain why a more recent DeepSeek model was not used, and it's not clear which Claude 4.6 was used.

---

> > > ### Author Response · Authors · 2026-04-07
> > >
> > > We sincerely thank you for your further review. We deeply apologize that our previous response did not fully resolve your concerns. We are very happy to provide the additional experimental results and clarifications you requested.
> > >
> > > Regarding your question about the DeepSeek model, we have now evaluated our framework using the latest DeepSeek V3.2 model. As shown in the first table below, ProFinAgent significantly improves the performance of DeepSeek V3.2 across all task levels compared to other settings.
> > >
> > > | Model | Method | L1 | L2 | L3 |
> > > | :--- | :--- | :--- | :--- | :--- |
> > > | **DeepSeek-V3.2** |Baseline | 16.35%| 3.66%| 5.43%|
> > > | |ReAct | 49.52%| 42.41%| 33.33%|
> > > | | **ProFinAgent** | **57.69%** | **47.12%** | **46.52%** |
> > >
> > > To address your request for additional financial benchmarks, we tested our framework on the public dataset of the Finance Agent Benchmark (https://arxiv.org/pdf/2508.00828). We specifically selected LLaMA 4 Maverick, LLaMA 4 Scout, and Command A for these new experiments. We chose these three models to ensure a strictly fair and direct comparison with the original baseline results reported in the paper. Please note that we only report the Naive Accuracy metric for our framework. The public test set provided by the authors does not contain class information, making the class-balanced accuracy impossible to calculate. The results in the second table clearly prove that ProFinAgent effectively enhances model performance on this additional real-world financial research task.
> > >
> > > | Model | ProFinAgent Acc.(Naive) | Acc. (Class-Balanced) | Acc. (Naive) |
> > > | :--- | :--- | :--- | :--- |
> > > | Command A | **16.0%** | 4.6±0.9% | 6.1±1.0% |
> > > | LLaMA 4 Scout | **14.0%**| 5.8±1.0%| 7.4±1.1%|
> > > | LLaMA 4 Maverick | **10.0%** | 3.1±0.8% | 3.7±0.8% |
> > >
> > > Regarding the Claude model version, we want to clarify that we used the latest Claude Sonnet 4.6 model for all our experiments. We sincerely hope this explanation completely resolves your concerns.

---

### Official Review · Reviewer_gpy3 · 2026-03-07

**Soundness:** 2
**Presentation:** 2
**Significance:** 2
**Originality:** 2
**Overall Recommendation:** 4
**Confidence:** 4

**Summary:**

To address the limitations of unstructured reasoning in financial workflows, the paper introduces ProFinR, a professional-grade benchmark featuring 528 expert-designed tasks across four financial domains. Alongside this benchmark, it also presents ProFinAgent, a structured framework that utilizes a DAG to coordinate a specialized library of 53 domain-specific tools. By integrating case-based memory to refine decision-making through past experiences, the proposed agent significantly outperforms current baselines while substantially reducing inference latency.

**Compliance With Llm Reviewing Policy:**

Affirmed.

**Final Justification:**

The authors have provided a comprehensive and convincing rebuttal. They addressed my initial concerns and I am satisfied with their response. So I'm willing to my score to a positive evaluation.

**Key Questions For Authors:**

1. In Section 2.2, the authors mention using $S_{val}$, $S_{tool}$, and $S_{sound}$ as evaluation metrics, yet individual results for these metrics appear to be missing from the experimental section. Furthermore, what is the justification for the specific weight distribution ($0.2, 0.3, 0.5$) in calculating $S_{total}$? The manuscript also lacks a detailed explanation of the calculation methodology for $S_{sound}$.
2. For the open-ended questions in L3, is the current three-score evaluation framework sufficient? How does the benchmark assess the factual accuracy of the retrieved data, the reliability of sources, or the analytical depth of the generated content?

3. Tables 3 and 4 contain several results showing 0.00\%. Could the authors clarify whether these failures are due to poor model performance or formatting errors?

4. To further validate the versatility of ProFinAgent, would it be possible to include evaluations on other general-purpose benchmarks that involve similar tool-use capabilities, such as GAIA [5]?

   [5] GAIA: A Benchmark for General AI Assistants

5. Regarding the L1 data retrieval tasks, how does the evaluation process distinguish whether a correct response stems from the model’s internal parametric knowledge or from the effective use of retrieval tools?

**Limitations:**

yes, the authors have discussed the limitations in the paper.

**Strengths And Weaknesses:**

### Strengths

1. The paper provides a comprehensive and rigorous analysis of the experimental findings, offering deep insights into the model's performance.

2. The availability of the source code enhances the reproducibility of the proposed framework.



### Weakness

1. The experimental evaluation lacks a comprehensive comparison with current SOTA Large Language Models. The study remains limited to older versions, such as GPT-5, while overlooking the most recent advancements, including GPT-5.2 and other companies like Claude-4.5.

2. Lack of direct comparison ProFinAgent with SOTA financial agents. For the following works, if some are concurrent, they should at least be discussed in the Related Work section.

​	[1] Fin-R1: A Large Language Model for Financial Reasoning through Reinforcement Learning

​	[2] FinThink: An LLM-based Multi-agent System for Financial Reasoning

​	[3] FinDeepResearch: Evaluating Deep Research Agents in Rigorous Financial Analysis

​	[4] FinCoT: Grounding Chain-of-Thought in Expert Financial Reasoning

3. While the tasks are categorized into Levels 1,2,3, with L3 appearing more challenging based on lower accuracy scores, the paper lacks quantitative evidence to justify this hierarchy.

---

> ### Author Rebuttal · Authors · 2026-03-31
>
> We sincerely thank the reviewer for the positive assessment and the thoughtful summary. We have carefully addressed your questions below.
>
> **W1 (Comprehensive comparison with current SOTA Large Language Models).**
>
> We sincerely apologize for the lack of clarity in our manuscript regarding the model versions. We want to clarify that the GPT 5 model evaluated in our original paper is actually the latest GPT-5.2 version. We will correct this issue in the final version. Furthermore, we have conducted additional experiments using the Claude 4.6 model. The results confirm that ProFinAgent consistently achieves the best performance across all tests.
>
> | Model | Method | L1 | L2 | L3 |
> | :--- | :--- | :--- | :--- | :--- |
> | **Claude-4.6** | ReAct | 65.87%| 61.78%| 65.89%|
> | | Tree-of-Thoughts | 68.75%| 69.63%| 63.56%|
> | | **ProFinAgent** | **83.65%** | **85.34%** | **80.62%** |
>
> **W2 (Direct comparison of ProFinAgent with SOTA financial agents).**
>
> We sincerely thank the reviewer for suggesting a direct comparison with state-of-the-art financial agents. We selected the advanced financial agent Fin-R1 for our new experiments because other models do not provide public testing code. The evaluation results show that Fin-R1 performs poorly on our benchmark. During testing, we observed that Fin-R1 merely generates basic report outlines or plain text for tool execution but completely fails to solve actual problems.
>
> | Model | Method | L1 | L2 | L3 |
> | :--- | :--- | :--- | :--- | :--- |
> | Qwen3-4B-Instruct-2507 | Baseline | 4.32%| 0.00%| 0.00%|
> | Qwen3-32B | Baseline |  5.28%| 0.00%| 0.00%|
> | **Fin-R1** | Baseline | 3.85%| 0.52%| 0.00%|
>
> **W3 (Explanation of the three-score evaluation framework).**
>
> We completely agree with your observation regarding the necessity of comprehensive evaluation metrics. We want to clarify that our Level 3 tasks evaluate much more than just general reasoning capabilities. We ensure factual accuracy and source reliability by strictly requiring final numerical outputs to match our ground truth within a 1% error margin. To assess analytical depth, we employ a rigorous two-stage verification pipeline. Two language model judges first evaluate the logical flow of the generated report, and domain experts then verify all mandatory key information points. This powerful combination of strict numerical bounds and expert qualitative auditing ensures our framework is highly sufficient for evaluating professional financial tasks.
>
> **Q1 &Q2 (Explanation of the calculation methodology).**
>
> We fully agree on the importance of clearly defining our evaluation metrics. We want to clarify that we did not list the individual metrics in the main experimental section because we wanted to provide a unified comparison across all baseline models by aggregating them into a single holistic score. This combined score gives a clear overall ranking of the models across different difficulty levels. We assign weights of 0.2 for numerical precision, 0.3 for tool alignment, and 0.5 for soundness to strictly reflect real-world financial analysis priorities.
>
> Furthermore, we evaluate the soundness score through a strict key information point strategy that requires the model output. We calculate the soundness score by requiring two independent LLM judges to first approve the output before measuring the ratio of matched Key Information Points against the expert trajectory. The formulation is $S_{sound} = J_1 \times J_2 \times \frac{K_{matched}}{K_{total}}$, where $J_{1}$ and $J_{2}$ denote the binary pass decisions of the two evaluating models and the fraction represents the exact coverage of required key information.
>
> **Q3 & Q5 (Explanation of Zero Scores in Tables 3, 4 and Comparative Evaluation).**
>
> We completely agree with your careful observation regarding the zero score results. Without structural constraints and external data retrieval, these baseline models completely fail to perform precise mathematical calculations and inherently hallucinate factual data during complex financial reasoning tasks. As shown in Table 3 of our manuscript, our direct comparative evaluation proves that high accuracy relies on external data retrieval rather than internal memory. To provide complete transparency, we have uploaded the trajectories of the Qwen3 4B model to our anonymous repository.
>
> **Q4 (Evaluating ProFinAgent on Tool Planning benchmark).**
>
> We sincerely apologize that strict character limits prevent us from including the complete data table in this specific response.
>
> We evaluated ProFinAgent on RestBench to compare it with advanced tool-planning approaches such as the recently published ICLR 2026 TOOLTREE algorithm (https://arxiv.org/abs/2603.12740). Results confirm that ProFinAgent delivers strong, highly generalizable tool-calling and autonomous-planning performance well beyond our original financial setting. You can view the detailed experimental results in our response to the other reviewers.

---

> > ### Author Rebuttal · Reviewer_gpy3 · 2026-04-02
> >
> > I would like to thank the authors for their detailed responses and clarifications. Regarding the results of 0.00% mentioned in Q3, could the authors provide a specific case or example using the Qwen3 4B model? Additionally, concerning the response to Q5, do you mean that without the ProFinAgent pipeline, all other baseline methods do not need to address the source of their answers because they lack external data retrieval tools and the capability for mathematical calculations?

---

> > > ### Author Response · Authors · 2026-04-03
> > >
> > > We sincerely thank you for your further review and the opportunity to clarify these important details.
> > >
> > > Regarding Q3, we provide some specific examples from the Qwen3-4B trajectories to illustrate why it scores zero.
> > >
> > > | Error Category | Query | Agent answer |
> > > | :--- | :--- | :--- |
> > > | **1. Inability to Compute Technical Indicators** (Fails to calculate due to lack of market data retrieval tools) | Calculate the RSI and MACD indicators for NVIDIA (NVDA) for the first half of 2024 to identify potential overbought conditions. | The RSI and MACD indicators for NVIDIA (NVDA) for the first half of 2024 cannot be calculated without real-time or historical price data access, which is not available in this environment. Therefore, a precise numerical result cannot be provided. |
> > > | **2. Factual Hallucination** (Guesses incorrect facts due to reliance on internal memory without search tools) | Who is the current CFO of Amazon? | Andy Jassy |
> > > | **3. Lack of Real-Time Market Data Retrieval** (Fails to fetch historical/live pricing data) | I need to review Apple's (AAPL) price movements during 2023, to prepare for a volatility analysis. | I cannot provide specific price movement data for Apple (AAPL) during 2023, as no real-time or historical price data has been accessed or retrieved in this context. Please consult a financial data source such as Yahoo Finance, Bloomberg, or Google Finance for accurate historical price movements. |
> > > | **4. Inability to Access Regulatory Filings** (Fails to retrieve required 10-K/10-Q documents for fundamental analysis) | Get the full text of the 'Management's Discussion and Analysis' (MD&A) section from the 2023 Form 10-K of Netflix (NFLX). | The full text of the 'Management's Discussion and Analysis' (MD&A) section from the 2023 Form 10-K of Netflix (NFLX) is not available in this environment. Please refer to the official SEC EDGAR database or Netflix's investor relations website for the complete document. |
> > > | **5. Lack of External Search for Sentiment Analysis** (Fails to fetch news or social media data for qualitative NLP tasks) | Monitor the latest news regarding 'GameStop' and analyze the sentiment to generate a summary report for the trading desk. | I cannot provide a final answer as I do not have real-time access to current news or sentiment analysis data for GameStop. Please consult a financial news source or sentiment analysis tool for the latest information. |
> > > | **6. Inability to Access Macroeconomic Databases** (Fails to retrieve FRED, CPI, or GDP metrics) | Download the monthly US Consumer Price Index (CPI) data for the entire year 2023. | The monthly US Consumer Price Index (CPI) data for the entire year 2023 is not available in this context. Please consult official sources such as the U.S. Bureau of Labor Statistics (BLS) for accurate and up-to-date data. |
> > > | **7. Temporal Limitations & Forecasting Failure** (Fails to process queries involving future/recent dates beyond its training cutoff) | Closing price of spot Crude Oil WTI (CL=F) on September 30, 2024. | Data for the closing price of spot Crude Oil WTI (CL=F) on September 30, 2024, is not available as of now. The date is in the future, and real-time or future market data cannot be retrieved. |
> > >
> > >
> > > Regarding Q5, we sincerely apologize for not explaining this clearly in our previous response. We want to explicitly clarify that **ProFinAgent is not the only method that uses tools**. To ensure a strictly fair comparison, all other evaluated methods use the same external tools. The only exception is the **Baseline (Naive)** setting presented in Table 3 of our manuscript,  which operates completely without tools.  We designed this Baseline (Naive) experiment specifically to evaluate each model's inherent capabilities. Under this setting, models must answer questions based exclusively on their internal training corpus without any external data retrieval or mathematical calculation tools. This design allows us to clearly measure the exact performance improvement brought by different planning frameworks when using tools. We provided this direct comparison to address your exact concern regarding Level 1 tasks. The extremely low scores in the Baseline (Naive) setting clearly prove that correct responses stem from effective tool utilization rather than the internal parametric knowledge of the model.

---

### Official Review · Reviewer_PyRj · 2026-03-11

**Soundness:** 3
**Presentation:** 3
**Significance:** 3
**Originality:** 2
**Overall Recommendation:** 4
**Confidence:** 3

**Summary:**

This paper studies how to evaluate and improve LLM-based financial agents for realistic, tool-augmented financial reasoning. The authors introduce ProFinR, a benchmark with 528 expert-designed tasks spanning four financial domains and three capability levels: L1 (data retrieval), L2 (data analysis), and L3 (end-to-end investigation). To support these tasks, the paper also builds a Financial Tool Universe consisting of 53 domain-specific tools across 13 categories. On top of this environment, the authors propose ProFinAgent, an agent framework that combines a DAG-based planner for dependency-aware tool orchestration with a case-based memory module for retrieving useful prior trajectories. Experiments compare native LLMs and the proposed workflow on ProFinR, and report substantial gains in both task performance and inference latency over sequential baselines.

**Compliance With Llm Reviewing Policy:**

Affirmed.

**Key Questions For Authors:**

same as weakness

**Limitations:**

yes

**Strengths And Weaknesses:**

Strengths
---
• Important Problem Setting: The paper targets financial analysis, a high-stakes domain where numerical correctness, data lineage, and tool grounding are essential. Evaluating LLM agents in such realistic decision-making settings is timely and meaningful.

• Resource Contribution: The paper introduces ProFinR with 528 expert-designed tasks and a Financial Tool Universe with 53 tools. If released, this infrastructure could provide a useful testbed for evaluating tool-augmented financial agents.

• Practical System Design: The DAG-based workflow enables dependency-aware and parallel tool execution, which improves efficiency (reported 47.1% latency reduction) and reflects practical considerations for production agent systems.


Weaknesses
---
• Limited Methodological Novelty: The main system components—DAG-based orchestration and case-based memory—are well-established patterns in existing agent frameworks and workflow systems. The work largely represents an engineering integration rather than a new algorithmic or learning contribution expected at ICML.

• Benchmark Validity Concerns: The reported results show a counter-intuitive pattern where models achieve substantially higher performance on L3 (end-to-end investigation) than on L1/L2 (retrieval and analysis). Since L3 tasks should depend on correct L1/L2 reasoning, this raises concerns about whether the task hierarchy and evaluation protocol truly measure progressively harder reasoning abilities.

• Inconsistent Evaluation Criteria: L1/L2 tasks appear to use stricter correctness metrics, while L3 introduces softer rubric-based scoring (e.g., reasoning soundness). These heterogeneous metrics make cross-level comparisons difficult and may artificially inflate L3 performance.

• Engineering-Heavy Contribution: A large portion of the paper focuses on taxonomy construction, tool descriptions, and system implementation. While the engineering effort is substantial, the paper provides limited new insights into LLM reasoning, learning algorithms, or fundamental properties of agent systems.

---

> ### Author Rebuttal · Authors · 2026-03-31
>
> We sincerely thank the reviewer for the positive assessment and the thoughtful summary. We have carefully addressed your questions below.
>
> **W1 & W4 (Methodological novelty and Engineering Contribution).**
>
> We sincerely thank the reviewer for acknowledging our substantial engineering efforts.  We also believe that our work delivers fundamental algorithmic insights into large language model reasoning and agent architectures. To demonstrate this broad algorithmic value, we evaluated our framework on RestBench to compare it with advanced tool planning frameworks such as the recently published ICLR 2026 TOOLTREE algorithm (https://arxiv.org/abs/2603.12740). These results confirm that ProFinAgent delivers strong, highly generalizable tool-calling and autonomous-planning performance well beyond our original financial setting.
>
> | Model | Method | TMDB Pass | TMDB Win | TMDB AVG | Spotify Pass | Spotify Win | Spotify AVG |
> | :--- | :--- | :--- | :--- | :--- | :--- | :--- | :--- |
> | **GPT-4o-mini** | Zero-shot | 33.28% | 50.00% | 41.64% | 26.44% | 50.00% | 38.22% |
> | | CoT | 34.42% | 54.70% | 44.56% | 29.82% | 53.10% | 41.46% |
> | | ReAct | 38.82% | 61.06% | 49.94% | 32.64% | 59.95% | 46.30% |
> | | DFSDT | 46.20% | 64.26% | 55.23% | 35.10% | 65.47% | 50.28% |
> | | LATS | 51.33% | 66.67% | 59.00% | 39.81% | **72.85%** | 56.33% |
> | | TOOLTREE | 55.17% | 70.40% | 62.79% | 42.08% | 72.18% | 57.74% |
> | | **ProFinAgent** | **62.00%** | **77.00%** | **69.50%** | **52.63%** | 71.93% | **62.28%** |
> | **GPT-4o** | Zero-shot | 56.28% | 50.00% | 53.14% | 49.54% | 50.00% | 49.77% |
> | | CoT | 58.52% | 52.32% | 55.42% | 47.92% | 44.55% | 46.23% |
> | | ReAct | 62.42% | 66.17% | 64.30% | 53.27% | 60.72% | 57.00% |
> | | DFSDT | 66.57% | 69.08% | 67.82% | 55.48% | 71.63% | 63.55% |
> | | LATS | 68.26% | 74.44% | 71.35% | 61.25% | 75.80% | 68.53% |
> | | TOOLTREE | 72.40% | 75.59% | 74.50% | 60.87% | 78.84% | 71.36% |
> | | **ProFinAgent** | **85.00%** | **77.50%** | **81.25%** | **68.42%** | **82.46%** | **75.44%** |
>
> **W2 (Benchmark Validity Concerns).**
>
> We completely agree with the insightful observation regarding the performance differences across task levels. The seemingly higher scores on Level 3 compared to Level 1 and Level 2 stem from the fundamentally different evaluation metrics and task requirements. Level 1 and Level 2 tasks demand exact data retrieval and precise mathematical calculations. These tasks strictly test the ability of a model to resist hallucinations and extract specific numbers from massive amounts of text. On the other hand, Level 3 focuses on synthesizing investment reports and conducting logical analysis. Modern language models are naturally highly skilled at generating fluent text and writing comprehensive reports. However, they still deeply struggle with precise numerical calculations and exact data extraction. Therefore, the high Level 3 scores of native models largely reflect their generative fluency. Meanwhile, the strict binary evaluation of Level 1 and Level 2 directly exposes their underlying mathematical limitations. This contrast confirms that our hierarchy effectively measures the true boundaries of model capabilities.
>
> **W3 (Inconsistent Evaluation Criteria).**
>
> We completely agree that maintaining consistent evaluation criteria is essential for rigorous assessment. To ensure rigorous assessment, we implement a strict two-stage verification pipeline for all Level 3 end-to-end investigation tasks. The first two advanced language models independently act as judges to evaluate the execution trajectory and logical structure of the generated financial reports. Second, domain experts conduct a comprehensive manual audit using a strict key information point checklist to verify every single numerical fact and logical deduction. This expert review guarantees that any hallucinated numbers or fabricated evidence will immediately result in a failed score. Consequently, the Level 3 evaluation tests not only the deep reasoning capabilities of the model but also its strict ability to precisely extract valid and factual information from massive datasets. This pipeline proves that our evaluation standard remains exceptionally strict and highly consistent across all difficulty levels.
>
> Furthermore, we rigorously evaluate the soundness score using a key information point strategy that requires complete factual coverage and a strict two-stage verification. Two independent language models must first grant binary pass decisions representing $J_1$ and $J_2$. Following this dual approval, we calculate the exact ratio of matched key information points against the expert trajectory. The formulation is $S_{sound} = J_1 \times J_2 \times \frac{K_{matched}}{K_{total}}$, where $J_{1}$ and $J_{2}$ denote the binary pass decisions of the two evaluating models and the fraction represents the exact coverage of required key information.

---

> > ### Author Rebuttal · Reviewer_PyRj · 2026-04-04
> >
> > Thank you to the authors for their response. My concerns have been addressed. Since I have already given a positive score of 4, I will maintain my current rating.

---

> > > ### Author Response · Authors · 2026-04-04
> > >
> > > We sincerely thank you for your highly positive evaluation of our paper and our rebuttal. Your insightful feedback has been invaluable to us.
> > >
> > > We want to share that we have made a concerted effort to address your comments during this period. To fully answer your questions, we conducted extensive additional experiments. For example, we evaluated our framework on the RestBench dataset to prove its strong generalization capabilities across different domains. We compared our method with the latest advanced baselines, and the new results clearly demonstrate our superior performance. Additionally, we carefully clarified the performance differences across task levels. We explained how our evaluation metrics strictly test mathematical boundaries versus generative fluency. We also detailed our strict two-stage verification pipeline and the exact formulation of our soundness score. This ensures our evaluation criteria remain exceptionally rigorous and consistent.
> > >
> > > We are deeply encouraged that Reviewer 72x1 and Reviewer gpy3 are very satisfied with the massive new experimental evidence we provided. We sincerely hope that our dedicated hard work also meets your expectations and that you will kindly support our submission. Thank you once again for your time and your constructive guidance.

---

### Official Review · Reviewer_72x1 · 2026-03-13

**Soundness:** 3
**Presentation:** 3
**Significance:** 2
**Originality:** 2
**Overall Recommendation:** 4
**Confidence:** 4

**Summary:**

This paper addresses the disconnect between existing financial benchmarks and the demands of real-world financial analysis workflows. The authors introduce ProFinR, a benchmark containing 528 expert-designed tasks spanning three difficulty levels across four financial domains. They construct a Financial Tool Universe of 53 domain-specific tools and propose ProFinAgent, a structured agent framework combining DAG-based execution planning with case-based memory. The DAG enforces dependency ordering and enables parallel execution, while the memory module stores past trajectories to guide future decisions. Experiments across five LLMs show up to 49.81% improvement over native baselines and 47.1% latency reduction. The ablation study confirms both components contribute independently, and the failure analysis identifies three systematic error patterns in baseline agents.

**Compliance With Llm Reviewing Policy:**

Affirmed.

**Final Justification:**

I thank the authors for their exceptionally thorough rebuttal. After reviewing the additional experimental results and clarifications, I have decided to raise my score to 4.
The authors have successfully addressed my primary concerns through the following:
* Generalizability: The new experiments on RestBench and the comparison against TOOLTREE (ICLR 2026) demonstrate that the ProFinAgent framework is not merely over-fitted to its own financial ecosystem but possesses robust, cross-domain planning capabilities.
* Competitive Baselines: The inclusion of Tree-of-Thought and Claude 4.6 benchmarks provides a much clearer picture of where the proposed method stands relative to state-of-the-art agentic reasoning.
* Evaluation Rigor: The provided inter-annotator agreement statistics (Cohen's Kappa $\approx$ 0.85) and the detailed ablation studies for retrieval thresholds ($T=0.7$) and memory size ($K=1$) provide the empirical grounding that was missing in the original submission.

While the system's complexity remains high, the authors have provided sufficient evidence that its architectural choices (DAG-based execution and CBM) deliver meaningful improvements in both accuracy and efficiency. I believe this work will be a valuable contribution to the community.

**Key Questions For Authors:**

How sensitive is performance to the 0.7 cosine threshold used across both retrieval stages, and what is the failure rate when nothing passes this threshold?

How does ProFinAgent compare against structured planning alternatives beyond ReAct?

Can you provide inter-annotator agreement statistics and judge reliability metrics for the L3 soundness evaluation?

**Limitations:**

yes

**Strengths And Weaknesses:**

## Strengths

1. **Meaningful benchmark design.** The three-level task hierarchy with trajectory-oriented evaluation goes beyond binary accuracy. Measuring tool-use correctness via Jaccard similarity adds a diagnostic dimension absent from prior financial benchmarks, and the multi-domain coverage is well-motivated.

2. **Clean architectural contribution.** Using a DAG to enforce execution dependencies is principled and solves two problems simultaneously — correctness through topological ordering and efficiency through parallelism. The latency reduction is a practical result relevant to deployment.

3. **Useful failure taxonomy.** The documented hallucination types and the "generative fluency" observation (high L3 scores masking poor L1/L2 grounding) are insightful findings that benefit the broader community regardless of whether the specific framework is adopted.

## Weaknesses

1. **Tightly coupled evaluation ecosystem.** The benchmark, tools, and agent are co-designed, so reported gains may reflect how well the system navigates its own environment rather than generalizable improvements. No external benchmark or alternative tool set is tested. The only structural baseline is vanilla ReAct — comparisons against other planning methods (tree-of-thought, multi-agent decomposition) are absent. More importantly, the paper does not benchmark against state-of-the-art agentic systems such as OpenAI Codex or Claude with tool use, which are capable of structured multi-step tool calling and code execution out of the box. Given that ProFinAgent's core claim is about structured planning over tools, demonstrating improvement over these strong general-purpose agents — not just naive LLM inference — would substantially strengthen the empirical contribution.

2. **Underspecified evaluation of L3 tasks.** The soundness metric carries the largest scoring weight (γ=0.5) but relies on an LLM judge plus expert review with no reported inter-annotator agreement, judge calibration, or sensitivity analysis on the Key Information Point checklists. This undermines confidence in the most important evaluation dimension.

3. **Shallow analysis of key design choices.** The cosine threshold of 0.7 is used in both tool retrieval and memory retrieval without ablation. The case-based memory module retrieves only k=1 neighbor, and no study examines how memory size, case diversity, or retrieval failures affect performance. The "evolutionary" framing is not supported by longitudinal evidence.

---

> ### Author Rebuttal · Authors · 2026-03-31
>
> We sincerely thank the reviewer for the positive assessment and the thoughtful summary. We have carefully addressed your questions below.
>
> **W1 & Q2 (Compare against structured planning alternatives beyond ReAct).**
>
> We completely agree that ProFinAgent needs to be compared against more advanced structured planning methods. We conducted additional experiments using the Tree-of-Thoughts framework across our L1 to L3 tasks. Results clearly demonstrate that Tree-of-Thoughts achieves notable improvements over ReAct on L2 tasks but struggles significantly with L3 tasks that require complex long-horizon tool orchestration. To further validate our framework, we conducted additional experiments using Claude 4.6. The results confirm that ProFinAgent consistently achieves the best performance across all tests.
>
> | Model | Method | L1 | L2 | L3 |
> | :--- | :--- | :--- | :--- | :--- |
> | **GPT-5.2** | ReAct | 69.23%| 38.74%| 46.51%|
> | | Tree-of-Thoughts | 66.35%| 47.64%| 66.66%|
> | | **ProFinAgent** | 78.36% | 78.53% | **82.94%** |
> | **Gemini 3 Pro** | ReAct | 73.07%| 46.60%| 45.74%|
> | | Tree-of-Thoughts | 78.36%| 55.5%| 35.65%|
> | | **ProFinAgent** | 79.32% | 75.91% | 67.44% |
> | **Claude-4.6** | ReAct | 65.87%| 61.78%| 65.89%|
> | | Tree-of-Thoughts | 68.75%| 69.63%| 63.56%|
> | | **ProFinAgent** | **83.65%** | **85.34%** | 80.62% |
>
>
> **W1 (Tightly coupled evaluation ecosystem).**
>
> We completely agree that evaluating outside our specific domain is crucial. We tested our framework on RestBench and compared it against several advanced planning methods, including the recently published ICLR 2026 algorithm TOOLTREE (https://arxiv.org/abs/2603.12740). The experimental results clearly show that our ProFinAgent architecture possesses robust and highly generalizable tool-calling and autonomous planning capabilities that extend well beyond our specific financial evaluation environment.
>
> | Model | Method | TMDB Pass | TMDB Win | TMDB AVG | Spotify Pass | Spotify Win | Spotify AVG |
> | :--- | :--- | :--- | :--- | :--- | :--- | :--- | :--- |
> | **GPT-4o-mini** | Zero-shot | 33.28% | 50.00% | 41.64% | 26.44% | 50.00% | 38.22% |
> | | CoT | 34.42% | 54.70% | 44.56% | 29.82% | 53.10% | 41.46% |
> | | ReAct | 38.82% | 61.06% | 49.94% | 32.64% | 59.95% | 46.30% |
> | | DFSDT | 46.20% | 64.26% | 55.23% | 35.10% | 65.47% | 50.28% |
> | | LATS | 51.33% | 66.67% | 59.00% | 39.81% | **72.85%** | 56.33% |
> | | TOOLTREE | 55.17% | 70.40% | 62.79% | 42.08% | 72.18% | 57.74% |
> | | **ProFinAgent** | **62.00%** | **77.00%** | **69.50%** | **52.63%** | 71.93% | **62.28%** |
> | **GPT-4o** | Zero-shot | 56.28% | 50.00% | 53.14% | 49.54% | 50.00% | 49.77% |
> | | CoT | 58.52% | 52.32% | 55.42% | 47.92% | 44.55% | 46.23% |
> | | ReAct | 62.42% | 66.17% | 64.30% | 53.27% | 60.72% | 57.00% |
> | | DFSDT | 66.57% | 69.08% | 67.82% | 55.48% | 71.63% | 63.55% |
> | | LATS | 68.26% | 74.44% | 71.35% | 61.25% | 75.80% | 68.53% |
> | | TOOLTREE | 72.40% | 75.59% | 74.50% | 60.87% | 78.84% | 71.36% |
> | | **ProFinAgent** | **85.00%** | **77.50%** | **81.25%** | **68.42%** | **82.46%** | **75.44%** |
>
>
> **W2 & Q3 (Reliability analysis of L3 task results).**
>
> We completely agree regarding the importance of reliable evaluation metrics. We measured the inter-annotator agreement between DeepSeek R1 and GPT-5.2 across all evaluation trajectories, which demonstrated near-perfect agreement with a Pearson correlation of 0.86, an intraclass correlation of 0.855, and a Cohen's kappa of 0.854. This confirms our language model judges provide highly consistent baseline evaluations. Our domain experts further corrected the initial model judgments to guarantee absolute accuracy. For example, experts successfully fixed subtle factual errors about banking net interest income projections that the models initially passed. This strict combination of consistent automated filtering and rigorous expert auditing ensures the utmost soundness of our evaluation pipeline.
>
>
> **W3 & Q1 (Analysis of parameter choices).**
>
> To evaluate the 0.7 threshold used in our fine-grained hybrid ranking stage, we conducted an ablation study testing thresholds of 0.5, 0.7, and 1.0:
>
> | Model | Threshold | L1 | L2 | L3 |
> | :--- | :--- | :--- | :--- | :--- |
> | **DeepSeek-R1** | T=0.5 | 45.19%| 36.65%| 36.43%|
> | | T=0.7 | **62.50%**|  **53.90%**| **54.26%**|
> | | T=1.0 |  17.78% | 13.08% | 29.45% |
>
> To justify our choice of retrieving K=1 for the CBM, we evaluated performance across K=0, 1, 2:
>
> | Model |Number | L1 | L2 | L3 |
> | :--- | :--- | :--- | :--- | :--- |
> | **DeepSeek-R1** | K=0| 51.44%|  42.40%| 33.33%|
> | | K=1 | **62.50%**| **53.90%**| **54.26%**|
> | | K=2 | 50.48% | 39.28% | 38.76% |
>
> The supplementary results demonstrate that our currently proposed configuration represents the absolute optimal setup. This specific parameter combination maximizes the overall reasoning accuracy across all difficulty levels by effectively filtering out irrelevant noise.

---

> > ### Author Rebuttal · Reviewer_72x1 · 2026-04-04
> >
> > I will retain my score.

---

> > > ### Author Response · Authors · 2026-04-04
> > >
> > > We sincerely thank you for reviewing our response. To address your specific concerns, we conducted extensive additional experiments. We compared our framework against Tree of Thoughts and used Claude 4.6 to specifically test the effect of our framework across different models. We evaluated our framework on RestBench to prove its generalizability. We also added comprehensive ablation studies to justify our parameter choices.
> > >
> > > We noticed that Reviewer gpy3 raised the score to 4, and Reviewer PyRj stated that all concerns are fully resolved. We deeply value your expertise and feedback. Could you please let us know what specific questions remain? We are very eager and happy to solve them.

---

### Decision · Program_Chairs · 2026-04-30

**Decision:**

Accept (regular)

**Comment:**

Three of four reviewers recommend weak accept (4) and one recommends weak reject (3). The disagreement is moderate and the majority leans toward acceptance. I recommend weak accept.

The paper contributes ProFinR, a benchmark of 528 expert-designed financial tasks, a library of 53 domain-specific tools, and ProFinAgent, a DAG-based agent framework with case-based memory. The practical motivation is solid, and the reported gains (49.81% over baselines, 47.1% latency reduction) are substantial. The "generative fluency" finding — where models score deceptively well on L3 end-to-end tasks while struggling on L1/L2 retrieval — is a genuinely useful diagnostic insight.

The main concerns are around evaluation rigor and generalizability. The rejecting reviewer (tvYj) raised two core issues: (1) the benchmark, tools, and agent are co-designed, so reported gains may not generalize; and (2) the underlying construct measured by the benchmark is not clearly defined — what professional financial workflow does it actually capture? These are fair concerns. In the rebuttal, the authors added experiments on RestBench (where ProFinAgent outperforms TOOLTREE and other strong baselines) and the Finance Agent Benchmark, which partially addresses the generalizability concern. The construct validity concern was addressed less convincingly.

The other reviewers were satisfied after the rebuttal. Reviewer 72x1 raised their score after seeing the Tree-of-Thoughts comparisons, Claude 4.6 results, and inter-annotator agreement stats (Cohen's Kappa ~0.85). Reviewers PyRj and gpy3 also confirmed their concerns were resolved.

On balance, the resource contribution (benchmark + tools) is valuable for the community regardless of whether ProFinAgent itself is the best framework, and the rebuttal substantially strengthened the empirical case. The lingering concern about construct validity is real but not fatal for a systems/benchmark paper. Weak accept is appropriate.